# Evolution of gut microbiota across honeybee species revealed by comparative metagenomics

Aiswarya Prasad[1], Asha D. Pallujam[2], Rajath Siddaganga [3],
Ashwin Suryanarayanan[3], Florent Mazel[1], Axel Brockmann[3], Sze H. Yek[2,4] &
Philipp Engel [1] ✉

Studying gut microbiota evolution across animals is crucial for understanding symbiotic interactions but is hampered by the lack of high-resolution genomic data. Honeybees, with their specialized gut microbiota and well-known ecology, offer an ideal system to study this evolution. Using shotgun metagenomics on 200 worker bees from five honeybee species, we recover thousands of metagenome-assembled genomes and identify several novel bacterial species. While microbial communities were mostly host-specific, we found both specialists and generalists, even among closely related bacterial species, with notable variation between honeybee hosts. Some bacterial generalists emerged host-specific only at the strain level, suggesting recent host switches. While we found some signal of co-diversification between hosts and symbionts, this was not more than expected by chance and was much less pronounced than what has been observed for gut bacteria of hominids and small mammals. Instead, symbiont gains, losses, and replacements emerged as important factors for honeybees. This highly dynamic evolution of the specialized honey bee gut microbiota has led to taxonomic and functional differences across hosts, such as the ability to degrade pollen-derived pectin. Our results provide new insights into the evolutionary processes that govern gut microbiota diversity across closely related hosts and uncover the functional potential of the previously underexplored gut microbiota of these important pollinators.

Animals harbor specialized microbial communities in their gut, which can influence health and disease susceptibility through diverse functional capabilities[1,2]. These communities are often host-specific, where the gut microbiota of individuals of the same host species are more similar in composition to each other than to those of different host species[3]. In some animal groups, it has also been observed that more closely related host species harbor more similar gut microbiota in composition, a pattern referred to as phylosymbiosis[4]. This host specificity at the community level arises from the host-restricted distribution of individual community members, which in turn is driven by microbial dispersal limitation of microbes and/or host-filtering processes[3,5].

Some gut bacterial clades—for example in termites[6] and hominids[7–11]—exhibit strong host specificity, with phylogenies that closely mirror those of their hosts. Such phylogenetic congruence (or

[1]Department of Fundamental Microbiology, University of Lausanne, Lausanne, Switzerland. [2]School of Science, Monash University Malaysia, Bandar Sunway, Malaysia. [3]National Centre for Biological Sciences, Tata Institute of Fundamental Research, Bengaluru, Karnataka, India. [4]Institute for Tropical Biology and Conservation, Universiti Malaysia Sabah, Kota Kinabalu, Malaysia. ✉e-mail: philipp.engel@unil.ch

co-phylogeny) can result from co-diversification (i.e., repeated co-speciation), in which hosts and symbionts undergo parallel speciation events, often driven by vertical inheritance from parent to offspring. However, current studies indicate that only a small fraction of the analyzed bacterial lineages show evidence of co-diversification. In contrast, numerous examples of generalist gut symbionts have also been documented[12,13]. This variation in the distribution of different gut bacteria suggests that these microbes do not follow the same evolutionary trajectory across various animal species. Indeed, a recent study[14] showed that the degree of host specificity differs across bacterial lineages in the animal gut, with both generalists and specialists present. Notably, this variation cannot be fully explained by microbial phenotype alone[15], highlighting a possible role of the host's ecology and environment in shaping these associations[3]. This raises the question of why generalist and specialist gut bacteria exist and whether specialist bacteria primarily evolved via repeated switches between hosts or through co-diversification with hosts.

Our current understanding of the evolution and diversification of gut microbial communities across hosts is limited by at least four technical and conceptual factors. (1) Previous studies have focused on highly diverse and variable systems across host individuals[16], making it challenging to understand whether gut communities change by acquisition of new members or by in situ evolution. (2) Significant differences in the relatedness and ecology of the compared hosts hinder the identification of the drivers of community change over evolutionary time scales and the prediction of their functional consequences. (3) Samples from different animals often come from different geographic regions, making geography an important confounding factor that could explain the host-restricted distribution observed for some gut symbionts[9,17]. (4) Most comparative studies are based on 16S rRNA gene analysis, which lacks the resolution to determine the distribution of closely related bacterial species, strains and functional gene content across hosts[18].

Comparative studies of the microbiota of closely related animals with similar phenotypes and geographic distribution at high taxonomic resolution are needed to overcome these limitations. The gut microbiota of honeybee workers is a promising model offering the combined advantage of a simple yet specialized microbial community shared among related host species with similar ecology, social behavior, and overlapping geographic ranges[19–21]. Prominent honeybee gut microbiota members include *Lactobacillus*, *Bombilactobacillus*, *Bifidobacterium*, and several proteobacteria. These bacterial lineages from honeybees are usually monophyletic but have diversified into host-specific species, several of which can coexist within the same host individual[22–24]. Thus, the predominant members of the honeybee gut microbiota have likely been acquired in a common ancestor of social bees and subsequently diversified in honeybees by separation into different host species and distinct ecological niches in the gut[25–27]. Co-diversification has been suggested as an important mode of evolution[21,27–30], but formal tests across the major community members have not yet been conducted[19]. So far, most comparative studies are based on either 16S rRNA gene amplicon sequencing with low genetic resolution or relatively few isolate genomes biased towards isolates from the managed Western honeybee *Apis mellifera*[27,30–36]. Both approaches provide limited quantitative insights into the distribution of divergent strains and species across hosts. Deep shotgun metagenomics allows the sequencing of most genomes in a given sample, characterization of the distribution of community members at high resolution (down to the strain level) across hosts, and assessment of the functional consequences of differences among hosts. However, this approach has only been used to study the Western honeybee (*Apis mellifera*)[21,30,37,38] and the Asian honeybee (*Apis cerana*)[39,40].

In this study, we applied shotgun metagenomics to individual worker bees of the five most prevalent species of honeybees described so far, and spanning all three major clades of honeybees, namely

Micrapis (dwarf honeybee): *A. florea* and *A. andreniformis*, Megapis (giant honeybee): *A. dorsata*, Apis: *A. mellifera* and *A. cerana*. While *A. mellifera* is globally distributed due to human introduction, the other honeybee species are mainly found across Asia[20]. All five honeybee species share several traits, such as a complex social lifestyle and a diet mainly consisting of processed pollen and nectar. However, they differ in certain key aspects, including body and colony size, nesting habit (open versus cavity nesting) and migrating ability[41].

From a total of 200 shotgun metagenomes from the gut of individual worker bees, sampled across Peninsular Malaysia and South India, we generated a comprehensive dataset of metagenome-assembled genomes (MAGs), expanding the catalog of the bacterial genomes available from these important pollinators and model for microbiome research. We reveal striking differences in host specificity across host and bacterial species and identify previously unrecognized but prevalent members of the bee gut microbiota that expand the functional repertoire of these communities in specific host species. Overall, our results show that the evolution of specialized gut microbiota-host interactions in honeybees has been shaped by symbiont loss and gain, and co-diversification.

## Results

### An expanded catalog of bacterial genomes from the gut of five honeybee species

We shotgun sequenced (35 M ± 19 M Illumina paired-end reads per sample) the hindguts of 200 individual worker bees from 40 different colonies of five honeybee species sampled in South India and various locations across peninsular Malaysia (Fig. 1A, Supplementary Data 1, Supplementary Fig. S1).

We recovered a total of 1959 high- and medium-quality metagenome-assembled genomes (MAGs, i.e., completeness score >50% and contamination <5%) (Supplementary Fig. S2), including 156 from *A. florea*, 163 from *A. andreniformis*, 869 from *A. dorsata*, 324 from *A. mellifera* and 413 from *A. cerana*) (Fig. 1B, Supplementary Data 2). The MAGs were discriminated into 150 species spanning 23 genera (prevalence > 10%) using 95% ANI (average nucleotide identity) clustering (Supplementary Fig. S3, Supplementary Fig. S4). A majority of these species belonged to genera previously reported to be predominant in the gut microbiota of honeybees, including *Lactobacillus* (formerly Firm5), *Bombilactobacillus* (formerly Firm4), *Bifidobacterium*, *Gilliamella*, *Snodgrassella*, *Frischella*, *Bartonella* and *Apibacter* (constituting 91/150 species clusters), but also lesser known genera such as *Dysgonomonas* (family Dysgonomonadaceae), *Saezia* (family Burkholderiaceae), *Pectinatus* (family Selenomonadaceae), *Entomomonas* (family Pseudomonadacea), *Pluralibacter* (family Enterobacteriaceae), WRHT01 (Desulfovibrionaceae family), CALYQQ01 (family Enterobacteriaceae) and CALYQJ01 (family Lactobacillaceae) (constituting 21/150 species). Around 47% of the identified bacterial species (71/150 species) have not been cultured to date, with many belonging to the predominant genera (41/71 species) and with genomic information currently lacking in public databases (Fig. 1B). The established collection of MAGs massively expands the catalog of bacterial genomes from the gut environment of these underexplored honeybee species.

### Most honeybee species harbor host-specific microbial communities in their gut

We assessed the taxonomic diversity of the gut microbiota across the five honeybee species using our collection of MAGs. Between 7 and 56 M reads from each sample mapped to these MAGs (Supplementary Data 3), providing sufficient depth of bacterial reads for MAG recovery and bacterial species detection from individual guts (Supplementary Fig. S5). Rarefaction curves show that our sampling effort from different colonies of each honeybee species (Fig. 1A) was sufficient to cover most of the species-level diversity in each honeybee species (Supplementary Fig. S6).

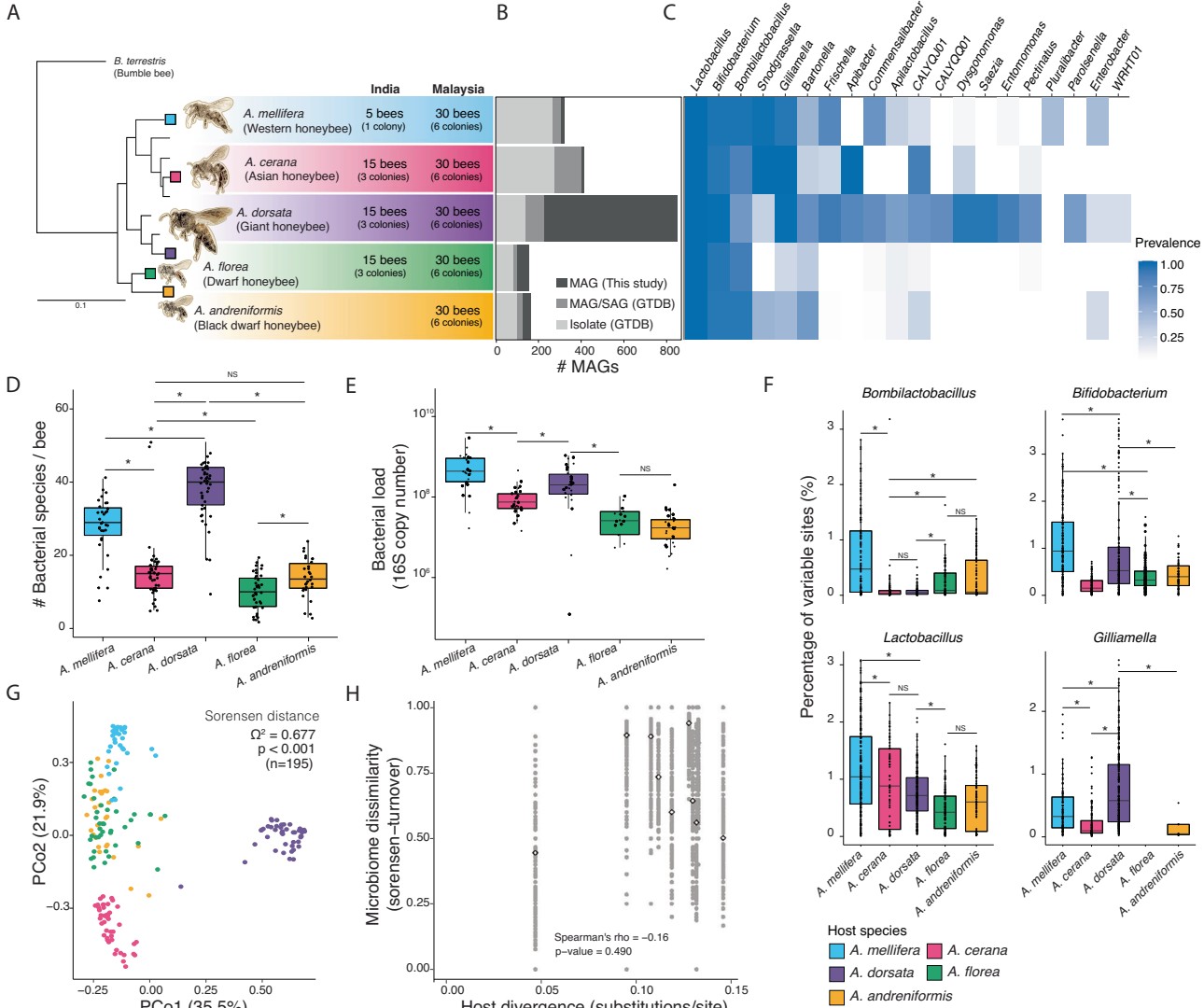

**Fig. 1 | Community-level patterns in the honeybee gut microbiome across taxonomic scales. A** Apis bee phylogeny with tips indicating extant species and the colored tips species included in this study. Pictures of bees (Kotthoff, U. et al.[99]) are to-scale relative to each other. Numbers and text next to species names indicate the sampling design. **B** Barplot showing the number of MAGs recovered from this study for which no isolate genomes of the same species (>95% ANI) were found in GTDB, those that were represented by MAGs or SAGs in GTDB and those that are similar to published genomes from isolated bacteria represented in GTDB. **C** Heatmap with the prevalence of each genus within each host species showing genera with prevalence >10% excluding the following three: *Klebsiella* (*K. pneumonia*)−10% in *A. mellifera*; Pantoea (*P. dispersa*)−17% in *A. mellifera*; *Melissococcus* (*M. plutonius*)−

13% in *A. andreniformis*). **D** Boxplot of the number of bacterial species in each honeybee gut across host species. **E** Boxplot of the number of 16S rRNA gene copies per 0.1µL of individual bee gut homogenate across host species. **F** Boxplot of the percentage of variable sites per individual of each host species shown for four most prevalent genera. Each point represents the percentage of variable sites found in each bacteria species of the genus within each individual of that host species. **G** PCoA plot based on the Sorensen distance across host species. **H** Scatter plot of microbiota dissimilarity (as the turnover component of Sorenson distance) as a function of host species divergence based on 16S + 12S rRNA (Ghonche-Golan, S. et al.[100]) with white points with thick black outlines marking the median of values for the comparisons of all pairs of samples from a given pair of host species.

Prevalence of the identified genera varied greatly across host species (Fig. 1C). *Lactobacillus*, *Bombilactobacillus*, and *Bifidobacterium* were consistently present across all five honeybee species (Fig. 1B). All other genera were only found in a subset of the five host species or occurred sporadically throughout. Notably, samples of *A. dorsata* contained several additional genera in high prevalence, such as *Dysgonomas* (90.9%), *Pectinatus* (70.5%), and *Saezia* (90.9%), which were rarely found in the other host species.

Comparison of the species-level bacterial diversity revealed that *A. dorsata* contained the most diverse microbiota, followed by *A. mellifera*, *A. cerana* and then the two dwarf honeybees *A. florea* and *A. andreniformis* (Fig. 1D). This pattern holds when considering only species of the shared genera (i.e., genera with a prevalence of at least

10% in each host), except that *A. dorsata* and *A. mellifera* were not significantly different from each other (Supplementary Fig. S7). Interestingly, the bacterial biomass per individual gut was the highest in *A. mellifera*, followed by *A. dorsata*, *A. cerana* and the dwarf bees, and hence, body size cannot necessarily explain the differences in alpha-diversity (Fig. 1E). Most bacterial species displayed significant strain level diversity as measured by the percentage of single nucleotide variants (SNVs) per bacterial species within each bee gut with the possible co-occurrence of several closely related strains (Supplementary Fig. S8). The Western honeybee *A. mellifera* hosted significantly more strains than other hosts for most bacterial species, while the dwarf honeybees harbored the least strain level diversity (Fig. 1F).

Taxonomic composition (Supplementary Figs. S9–10) was host-specific at the species level as suggested by β-diversity analyses (PERMANOVA Sorenen distance $\Omega^2 = 0.6777$, $p = 0.001$, $N = 195$), but with no distinction between the two dwarf honeybee species (Fig. 1G) and no influence of geography ($\Omega^2 = 0.009$, $p = 0.086$, $N = 131$ excluding two host species—*A. andreniformis* and *A. mellifera*—sampled unequally across locations) (Supplementary Fig. S11). We found no evidence for phylosymbiosis, as there was no correlation between bacterial community distance and host divergence (Spearman's $p$-value = 0.49, mean $q$-value = 0.619 ± 0.204 across 1000 mantel tests, Fig. 1H). For example, *A. andreniformis* and *A. mellifera* are the most distant host species but have much smaller distances in bacterial community composition than other pairs of host species.

## Host specificity of individual community members varies across bacteria and host species

Specificity at the community level emerges because of the host-specific distribution of individual community members[14]. We calculated the specificity per bacterial species based on Rohde's Index[42], which considers the prevalence of each community member across samples from different hosts (Fig. 2A, Supplementary Data 4).

Overall, we found large differences in the specificity of individual bacterial species both within and between genera. For example, the genus *Bombilactobacillus* mainly consisted of generalist species found across all five honeybee species, while the genera *Lactobacillus* or *Bifidobacterium* contained specialists and generalists (Fig. 2A). Some bacterial species showed an intermediate level of specificity as they were shared only among a subset of the five host species and were more prevalent in one host than another (e.g., one of the *Snodgrassella* species). Few shared genera consisted of only specialist species (*Apibacter* and *Commensalibacter*). Further, there was a clear difference in the specificity of bacterial species between hosts: the microbiota of *A. dorsata* and *A. mellifera* were composed of both generalist and specialist species, while the microbiota of the other smaller honeybee species (*A. cerana*, *A. florea*, and *A. andreniformis*) was primarily composed of generalist species (Fig. 2A).

Bacterial species may be shared between host species but exhibit host specificity at the strain level. To look for such strain-level host specificity, we measured the extent of shared SNVs between samples for each bacterial species shared between two or more host species using popANI[43]. We found both bacterial species that segregate and species that do not segregate by host species in terms of strain-level composition (Fig. 2B, C). This pattern was observed across all analyzed genera (Fig. 2B, C). Moreover, in some cases, we also found that samples segregated by sampling location (Fig. 2B, C). Overall, 75% of the species were host-specific at either strain or species level.

## Host switches and species gains and losses interrupted the host-specific evolution of gut microbiota members of honeybees

Co-diversification has been described in other host-associated microbiomes as an important driver of host specificity patterns[7,10,11]. To test for evidence of co-diversification in the honeybee gut microbiota, we measured congruency between host and bacterial phylogenies (or co-phylogenetic patterns). In case of strict co-diversification one would expect the bacterial and host phylogeny to mirror each other at all nodes (see Fig. 3A for a scheme). We constructed core genome phylogenies for the nine most prevalent genera using the collection of all high- and medium-quality MAGs assembled from the five honeybee species, including several MAGs from each bacterial species within the genus. We used genus-level phylogenies to test for co-diversification because divergence time within genera was roughly equivalent to divergence times in the *Apis* genus (see "Methods"). The resulting trees are depicted in Fig. 3B, but can also be inspected interactively online [https://itol.embl.de/shared/Aiswarya]. Most honeybee-derived MAGs from a given bacterial genus formed clades closely related to those

from bumble bees and stingless bees, but segregated into distinct species-level clusters based on ANI (Fig. 3B and Supplementary Fig. S3). In agreement with the observed differences in host specificity, 24–47% of these species-level clades were exclusive to a single host species, while the remainder included MAGs from multiple hosts. To test for congruency between host and bacterial phylogenies, we used the Hommola cospeciation test, a generalization of the Mantel test for host-parasite interaction graphs[44]. We applied this test to all nodes containing at least 7 different MAGs from three or more host species. The Hommola test detects phylogenetic congruence driven by two main factors: (i) clustering of bacterial taxa (i.e., tree tips) by host species, and (ii) topological congruence of internal branches between bacterial and host phylogeny (see scheme in Fig. 3A). Only the latter provides evidence for true co-diversification across host species. However, the former can also produce low p-values and elevated R values, as noted in previous studies[7,45] and demonstrated by our evaluation of various thresholds (see Supplementary Fig. S12, Supplementary Data 5–6, and "Methods"). To minimize false positives, we adopted stringent significance criteria (R > 0.75 and p < 0.01), consistent with the thresholds used in prior work[7]. We then carried out second-order permutation tests ($n = 100$) with shuffled tip labels to account for the rate of false discoveries as described in previously published work[45] (see "Methods" for more details and an illustration of the approach). Overall, this analysis identified significant phylogenetic congruence between hosts and symbionts in only 13 of the 192 tested nodes/subtrees (Fig. 3B and online[46] using iTOL v6 [https://itol.embl.de/shared/Aiswarya]), which was no more than expected by chance in any of the nine tested genera as determined by our permutation tests (Fig. 3C, Supplementary Data 5–6). Visual inspection confirmed that a subset of the significant subtrees exhibited topological congruence with the host phylogeny, consistent with co-diversification—for example, in subtrees of *Bifidobacterium* (Supplementary Fig. S13A, B), *Commensalibacter* (Supplementary Fig. S13C), or *Snodgrassella* (Supplementary Fig. S13D). In contrast, other significant subtrees, lacked topological congruence with the host tree (e.g., *Frischella* subtree in Supplementary Fig. S13E, or *Bombilactobacillus* subtree in Supplementary Fig. S13F), or showed internal branch lengths inconsistent with host divergence times (e.g., *Lactobacillus* subtree in Supplementary Fig. S13G, *Bombilactobacillus* subtree in Supplementary Fig. S13H, or *Bartonella* subtree in Supplementary Fig. S13I). Notably, none of the significant subtrees included symbiont clades from all five honeybee species, suggesting that these co-diversifying lineages were either lost in some hosts or not acquired in the common ancestor of the five honeybee species but in later stages.

In summary, we find relatively few nodes with clear evidence of co-diversification. Instead, as seen in Fig. 3B, deep-branching host-specific clades are frequently interrupted by generalist clades, indicating recent host switches. Likewise, deep-branching generalist clades—some of which exhibit host specificity at the strain level—often neighbor host-specific clades indicating recent host specialization that likely occurred after host divergence. Together, these patterns reflect a dynamic evolutionary history shaped by host switching and secondary specialization.

## Functional differences in the gut microbiota across honeybees

To test whether the host specificity observed at the taxonomic level translates into functional host specificity of the honeybee gut microbiota, we performed a functional analysis of the MAGs. Of 4,759,622 prokaryotic ORFs in our MAG database, more than half (2,685,895 ORFs) were assigned a KO (KEGG Ortholog) identifier. The rarefaction curve analysis of these KOs indicated that most of the functional diversity across the analyzed samples was recovered with our MAG database (Supplementary Fig. S14). Overall, the number of KOs per sample followed a similar trend as the taxonomic diversity, with *A. dorsata* having the most and *A. florea* having the fewest KOs (Fig. 4A).

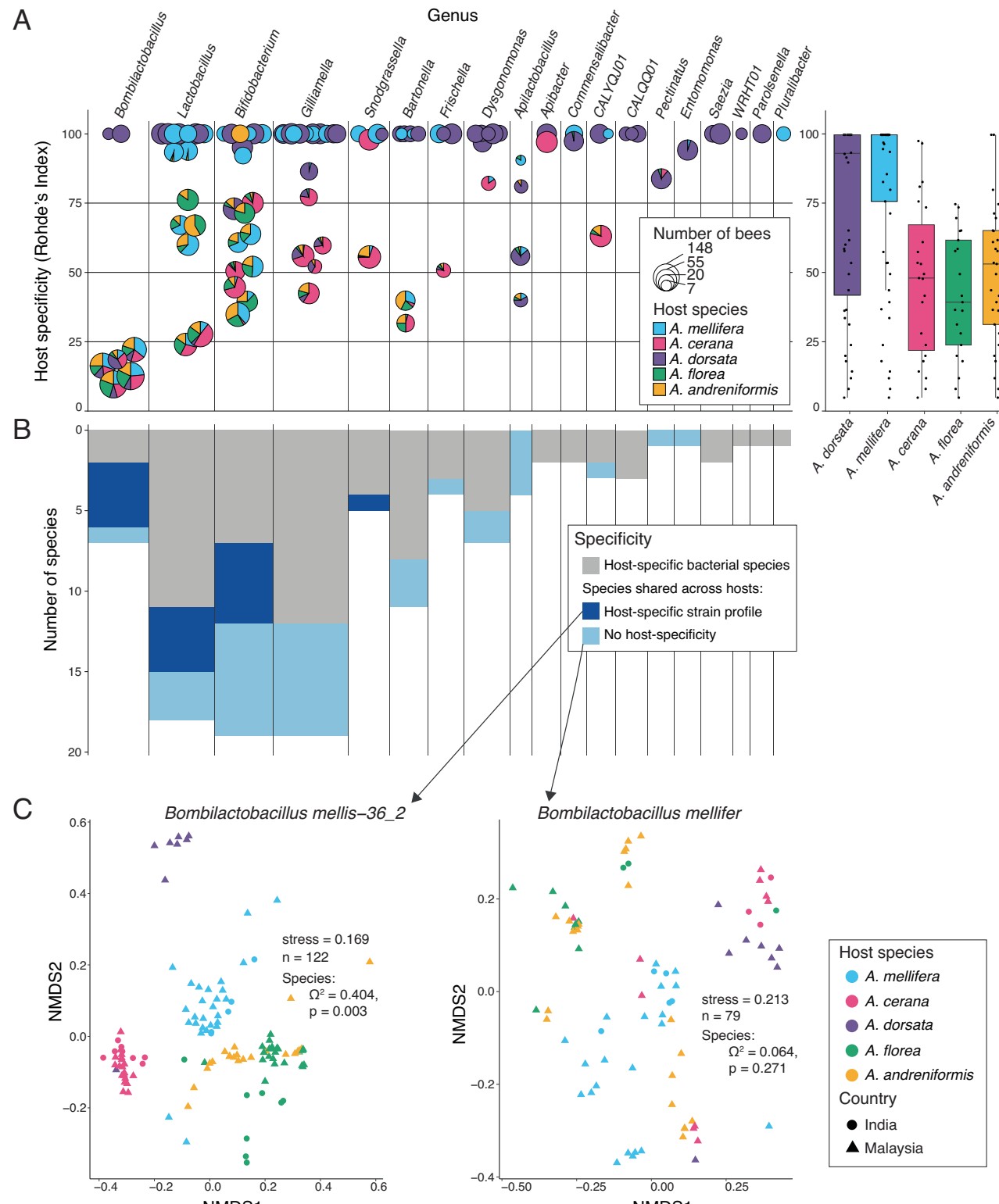

**Fig. 2 | Host-specificity of individual gut microbial lineages across honeybee hosts. A** Scatter-pie plot of Rohde's index of specificity with each circle representing a bacterial species with the size representing the number of bees in which the species was found, and the pie chart showing the proportion of samples belonging to each honeybee host. The adjacent boxplot shows Rohde's specificity index for each species found in each honeybee host. **B** Distribution of species richness per genus across three categories of specificity levels: host-specific at the species level (gray), host-specific at the strain level, but the species is detected across hosts (dark blue) or not specific even at the strain level (light blue). **C** NMDS plots from a strain level generalist and specialist bacterial species are included as an example. The Jaccard dissimilarity was calculated in terms of strain level composition (popANI) tested using PERMANOVA (significant if the *p*-value is <0.05). Only bacterial species with enough coverage to recover SNPs in at least five samples spanning at least two host species were included.

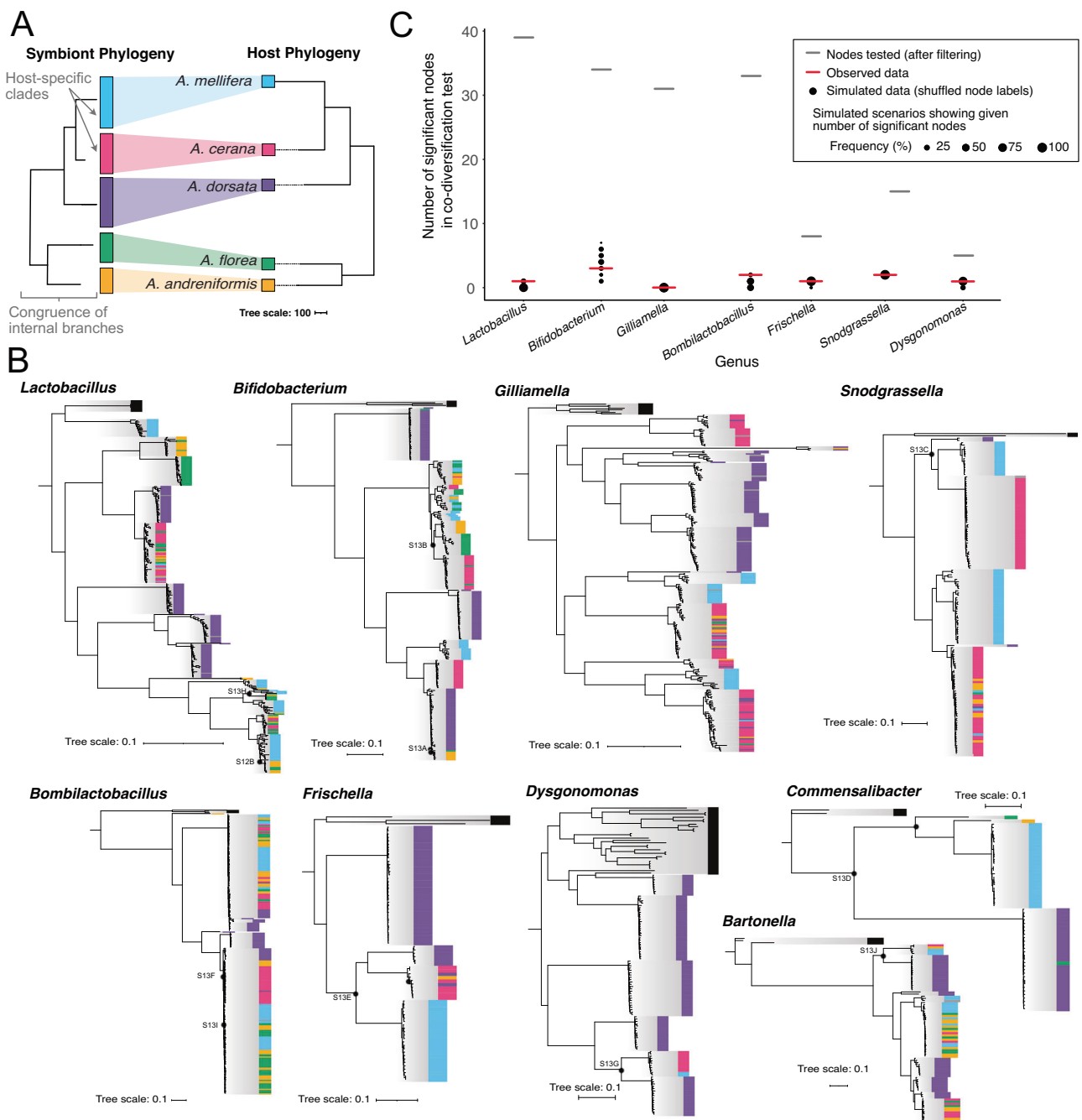

**Fig. 3 | Bacterial genus-level core genome phylogenies of MAGs. A** A tanglegram of a theoretical phylogeny of symbionts (left) showing perfect congruence with the host tree of Apis (right). The host tree is based on actual data and to scale according to source (tree data adapted[SS]). The scale bar is the inferred number of nucleotide substitutions. Signal for topologcial congruence between symbiont and host phylogeny in the Hommola test of co-phylogeny can result from two effects, as indicated: (i) congruence at (internal) branches separating clades of symbionts from different hosts, and (ii) clustering of symbionts into host-specific clades (without congruence at internal nodes separating them). **B** Core genome phylogenies of MAGs from the nine most prevalent genera found in more than one host species. Colors at the leaf tip indicate the host species from which the MAG was recovered and solid gray boxes depict the set of genomes included to represent outgroup species of the genus from environments other than honeybee guts. The scale bar represents 0.1 amino-acid substitutions per site. The phylogenies are maximum likelihood trees inferred by IQ-TREE (see "Methods"). Black filled circles indicate subtrees of nodes that were significant in the Hommola test for co-phylogeny. These subtrees are further illustrated in Supplementary Figs. S12–S13. **C** Scatter plot representing the number of nodes significant (under strict threshold) for co-diversification in the actual node-by-node comparison (red line) compared with that in the second-order permutation test comparisons ($N = 100$) with host tree node labels shuffled. The other short horizontal line (gray) denotes the number of nodes compared after excluding those that were filtered (did not have at least 7 MAGs from at least 3 different host species).

Multivariate analysis of normalized KO counts (and the Robust Aitchison distance) showed that the functional composition of the gut microbiota of five honeybee species was distinct, even between *A. florea* and *A. andreniformis*, which overlapped in composition at the same taxonomic level ($\Omega^2 = 0.148$, $p = 0.001$) (Fig. 4B). To understand which bacteria drive these differences, we performed a similar analysis using counts of all KO families in each high- and medium-quality MAG across all samples. As expected, MAGs of bacterial species from the same

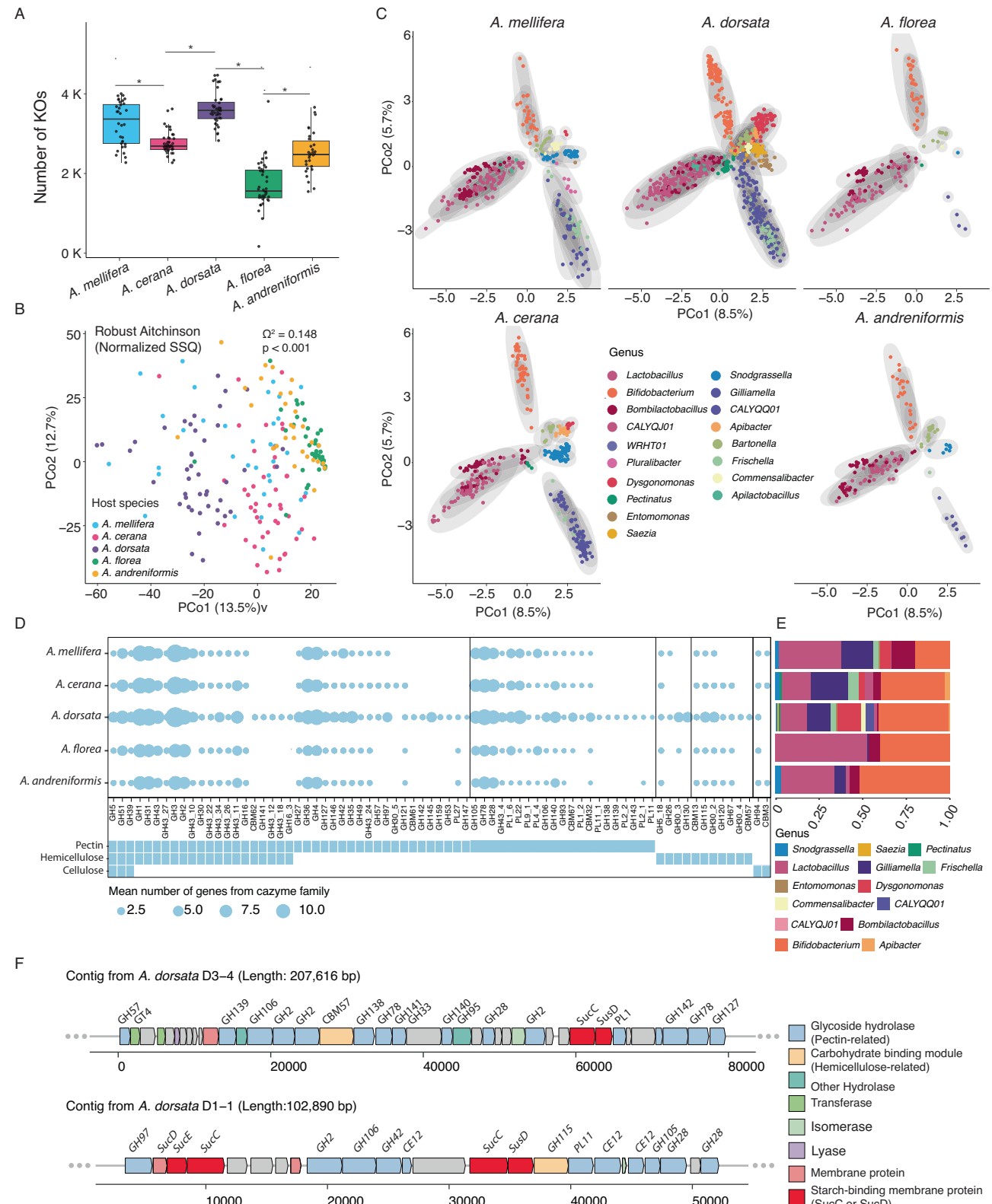

genus clustered across all host species, suggesting that they occupy similar functional niches and that the overall functional capacity of the bee microbiota is conserved across hosts (Fig. 4C). Yet, there were some important differences. Some of the genera only found in *A. dorsata* formed distinct clusters (e.g., *Dysgonomonas*), suggesting that these bacteria occupy unique functional niches in the microbiota of this host species. Other genera (e.g., *Saezia* and *Entomomonas*) clustered with community members prevalent in other honeybee

species (e.g., *Snodgrassella*) but absent from *A. dorsata* (Fig. 4C). Closer inspection of their metabolic profiles indicated that both *Saezia* and *Entomomonas* like *Snodgrassella* lack major carbohydrate metabolism pathways (Supplementary Fig. S15), suggesting potential functional replacements. Finally, although MAGs from the same genus mostly overlapped in terms of their functional profile, we found some cases where MAGs of the same bacterial genus clustered by honeybee species from which they were recovered, suggesting host-specific

**Fig. 4 | Functional potential of honeybee gut microbiota across host species.**
**A** Boxplot of the number of KEGG ortholog (KO) families detected per individual of each host species Wilcox test ($p < 0.05$) indicated on the plot. **B** PCoA of KO family composition across all samples, visualized using the Robust Aitchison distance on a matrix of reads per kilobase per million (RPKM) summed across genes belonging to the respective KO family. Each point on the plot represents an individual honeybee sample and the color the host species identity of the sample. PERMANOVA test indicated on the plot tests the effect of host species identity. **C** PCoA of all KO families from each MAG using Robust-Aitchison distance on the RPKM matrix of KOs. Each point represents a bacterial MAG colored by the bacterial genus it belongs to. MAGs obtained from different host species were separated across panels for clearer visualization. The axes across the five panels have been maintained to the same limits. So, the panels can be superposed. **D** Dotplot of the mean number of genes from each CAZyme family with a potential role in pectin, hemicellulose, or cellulose degradation per individual of each host species. **E** Barplot of the proportion of genes from CAZyme families contributed by each bacterial genus. **F** Loci within Dysgonomonas MAGs encoding putative polysaccharide utilization loci (PULs).

differences in the functional repertoire at the species-level (and potentially, adaptation) within these genera (Supplementary Fig. S16).

Since pollen-derived glycan (i.e., pectin and hemicellulose) breakdown has been described as one of the characteristic catabolic functions of the gut microbiota of *A. mellifera*[21,47], we specifically looked at the Carbohydrate-active enzyme (CAZyme) repertoire of the microbiota across the five host species. We detected 19,562 CAZyme families among the prokaryotic ORFs in our MAG database. Overall, the composition of these gene families was host-specific (Supplementary Fig. S17). While all five honeybee species harbored a similar set of CAZymes for pectin, hemicellulose, and cellulose degradation, *A. dorsata* contained a more extensive and diverse set of such gene families, especially many of which are involved in pectin degradation (Fig. 4D). To understand which bacteria drove these differences, we considered the number of genes belonging to CAZyme families from each bacterial species. Previously recognized members contributed most families to degrade pollen-derived glycans, specifically *Gilliamella* and *Bifidobacterium*[21,47] (Fig. 4E), except in the giant honeybee *A. dorsata*, where the genus *Dysgonomonas* also contributed key enzyme families involved in pectin degradation such as PL1_2, PL11_1, GH28, GH78, GH138, or CE12 (Fig. 4F, Supplementary Fig. S18). Interestingly, the CAZyme genes of *Dysgonomonas* were often organized in large genomic islands reminiscent of specialized PULs (polysaccharide utilization loci) found in gut symbionts of mammals[48], suggesting that this community member contributes to the capability of this particular honeybee species to degrade plant-derived fibers (Fig. 4F). Moreover, the fact that the MAGs of *Dysgonomonas* found in *Apis dorsata* form deep-branching clades, only distantly related to species isolated from cockroaches[49] and human stool[50], suggests host specialization.

## Discussion

Many animals harbor host-specific bacterial communities in their gut[3–5,14]. However, it is still unclear whether and why host specificity varies among closely related host species or bacterial lineages, and which evolutionary processes underlie the observed patterns of specificity and their functional consequences. Our comparative shotgun metagenomic analysis of the gut microbiota of five honeybee species provides novel insights into these questions. It advances our understanding of the evolutionary processes that govern gut microbiota composition across closely related animal species.

We find that the community composition of the gut microbiota of three (*A. mellifera*, *A. cerana*, *A. dorsata*) of the five analyzed honeybee species is host-specific. In contrast, the gut microbiota of the two dwarf honeybee species, *A. florea* and *A. andreniformis*, were indistinguishable from each other, suggesting that host specificity can vary between closely related host species.

Our genome-level data, coupled with our broad sampling design, allowed us to move beyond community-level patterns and to identify host specificity of individual community members. We found that the vast majority of the analyzed community members were host-specific, but also that host specificity emerges at different taxonomic levels, for example, at the genus (such as *Saezia* and *Dysgonomonas* in *A. dorsata*), species (such as many *Gilliamella*, *Snodgrassella* and *Lactobacillus* species), and strain level (such as *Bombilactobacillus mellis-36_2*). Considering that all of these bacteria belong to clades specific to social

bees, this suggests that the host-specific association between different gut symbionts and their honeybee hosts has been established across different time points in their evolutionary history. These findings also highlight that patterns of host-specific associations can be overlooked without strain-level/genomic resolution.

Another key finding of our analysis is that the degree of host specificity varies across bacteria (i.e., there are generalist and specialist species). This has also been observed in other systems[14,15,51] and linked to bacterial traits, such as aerotolerance and spore-formation, that affect dispersal ability[14,52–55]. However, unlike previous studies, we found unexpectedly high variability in host specificity even among closely related bacteria, i.e., species belonging to the same genus. This suggests that conserved microbial traits alone cannot fully account for variation in specificity patterns. Instead, our results point to the identity of the host species as a key determinant of bacterial host specificity. Most of the highly host-specific bacterial species were found in *A. dorsata* and *A. mellifera*, while the communities of *A. cerana* and the two dwarf honeybees were dominated by shared bacterial species. *A. mellifera* and *A. dorsata* guts may harbor more specific ecological niches[56] (a narrow range of conditions or resources that an organism can exploit) than the other bee species, facilitating the colonization and maintenance of additional 'specialist' bacteria. This is consistent with the observation that *A. cerana* and the two dwarf honeybees harbor significantly fewer bacterial species per individual bee than *A. dorsata* and *A. mellifera*. Several different factors could contribute to these patterns. Ecological niches in the host gut can vary even among closely related host species due to differences in host physiology, diet, or gut structure. For example, the host-specific colonization of *S. alvi* of *A. mellifera* may be supported by specific organic acids secreted by the host into the gut lumen and utilized by this bacterium[57]. Slight differences in dietary preferences, e.g., the amount or type of pollen, may allow different combinations of species to co-exist in one host species compared to another[58]. Variation in body and colony size between *Apis* species may determine the carrying capacity and hence bacterial diversity in the gut, as proposed previously[27,39]. Another possibility is that host species differ in the degree of control they exert over their gut microbiota—potentially through mechanisms such as immune regulation, as suggested in other animals[59]. In general, the molecular and ecological drivers of host specificity remain poorly understood. However, the host-specific clades and those showing evidence of recent host switches identified in this study offer promising targets for future research into these processes.

A central objective of our study was to uncover the evolutionary processes shaping the above described structure and distribution of gut symbiont diversity in honeybees. While it is generally assumed that the honeybee microbiota has co-diversified with their hosts—with occasional interruptions—previous studies have not applied recent analytical frameworks that allow comparison across gut microbiome systems to rigorously test these assumptions[25,27,29,60]. Using our analytical approach, we found limited evidence for strict co-diversification between gut bacteria and their hosts across the five analyzed *Apis* species. Although we applied relatively stringent thresholds (i.e., high R values) to detect signals of co-diversification, the second-order permutation analysis revealed that the number of significant nodes did

not exceed what would be expected by chance independent of the chosen R-value cutoff. Rather than supporting widespread co-diversification between gut bacteria and honeybee hosts, our findings highlight the importance of symbiont gain and loss, as well as host switching, in shaping these specialized gut communities. Several lines of evidence support this conclusion. First, only a small proportion of nodes in the genus-level phylogenies showed significant congruence with the host phylogeny—pattern that would be expected in case of co-diversification. Second, some bacterial genera that are abundant in certain honeybee species are completely or nearly absent in others, suggesting the gain or loss of entire genera during host evolution—an observation also reported for the microbiota of closely related stingless bees[61]. Third, even among genera shared across all five honey bee species, the subtrees displaying strong signals of co-diversification never encompassed all hosts, implying lineage-specific acquisition or loss of these symbionts. Fourth, several species-level clades (24–47% of all species-level clades) include MAGs from multiple host species, consistent with recent host switching or ongoing interspecies transmission. Fifth, internal branch lengths varied substantially among splits of symbionts associated with the same hosts, suggesting that these divergence events occurred at different points in evolutionary time, although they could also reflect differences in evolutionary rates.

These results appear to contradict previous studies that reported evidence for co-diversification between bee gut symbionts and their hosts[25,27,28,62]. This discrepancy may stem from the more comprehensive approach of our study, which incorporates improved design and analytical methods, allowing us to detect patterns that may have been overlooked in previous work. However, even the earlier studies acknowledged that host switches, symbiont gains, and losses must have occurred, as evidenced by the imperfect congruence between host and bacterial phylogenies. For example, one study that applied statistical methods and found evidence for co-diversification still estimated that the rate of co-speciation was about half the rate of symbiont switches[28]. Furthermore, this study focused primarily on bumble bee bacteria from the genera *Snodgrassella* and *Gilliamella*. It is possible that the signal for co-diversification in gut symbionts of bumble bees may be stronger compared to those from honeybees. This hypothesis should be tested in future studies using more comparable approaches, including a larger number of host species, and complementing analysis based on metagenomes with isolate genome data. Our results also markedly contrast with co-diversification findings in mammals, specifically hominids[6–11] and rodents[63]. Those studies, using a similar approach, found that the ratio of the number of significantly congruent clades observed and expected only by chance was one order of magnitude higher than in our dataset (-12 vs. -1.2). However, even in hominids, only 16–36%[7,9,11] of all tested bacterial clades showed statistically significant signals of co-diversification, suggesting that the evolution of only a minority of the microbiota is driven by co-diversification.

While the social lifestyle of bees likely facilitates within-host transmission and could, over evolutionary timescales, promote co-diversification, ecological opportunities for microbial spillover may be relatively common—perhaps due to the sympatric distribution of *Apis* species in Asia. As a result, long-standing associations between specific hosts and their gut symbionts may occasionally be disrupted by the invasion of competing microbes from related host species, thereby limiting the potential for symbionts to consistently co-diversify with their hosts across species boundaries. Furthermore, since all five honeybee species have similar dietary preferences, gut structure, and physiology and lack an adaptive immune system, the barrier to colonization between native and non-native bacteria may be low. This is supported by experiments with microbiota-free bees, which showed that bacterial strains isolated from honeybees can colonize a non-native honeybee species and sometimes even outcompete native bacterial strains[26,27].

Finally, our study higlights that the host-specific composition of the microbiota in these honeybee species translates into meaningful functional differences. In particular, both the presence of different variants of conserved members and the gain of novel taxa with distinct functional capabilities significantly influenced microbiome function. For example, *Apis dorsata* harbors honey bee-specific lineages of *Dysgonomonas*—a genus of polysaccharolytic Bacteroidetes also found in the guts of cockroaches, termites, and humans[49,50,64]—which substantially expanded the repertoire of pectin and hemicellulose degradation genes in the gut microbiota of *A. dorsata* without necessarily replacing other gut symbiont. *A. dorsata* may be distinct in its broad exposure to more diverse microbes compared to other honeybees, given its highly migratory nature[65]. Interestingly, differences in diet may not explain differences in microbiome diversity, considering that *A. dorsata* tends to forage on fewer mass-flowering plants[66], and plant-pollinator network analysis suggests that *A.florea*, *A. cerana*, and *A. dorsata* show substantial overlap in the flowers they visit[67]. In summary, the honeybee gut microbial community shows a highly dynamic evolution despite the presence of a few conserved lineages.

Our results lay the foundation for more detailed functional characterization of diverse bacterial gut symbionts that were previously unrecognized as important members of the honeybee gut microbiota. This work substantially broadens our understanding of honeybee-associated microbiota, which has so far been based primarily on studies of the managed Western honeybee. More broadly, it provides new insights into host specificity in gut microbiota-host interactions, demonstrating that a dynamic evolutionary history can critically shape microbiome functions across divergent animal hosts.

## Methods
### Sampling and storage
In Malaysia, sampling involved the collection of honeybees from reserved forest areas. The permit application was approved by the Perak State Forestry Department to collect the samples from Temenggor forest with the approval number: JH/100 Jld.31(46). Transfer of biological material was carried out under the Access and benefit-sharing (ABS) act, issued by Ministry of Natural Resources and Environmental Sustainability (NRES), Malaysia, April 2019. Reference number: 676216. In India, honeybee sampling was conducted from colonies on the campus of the National Centre for Biological Sciences – TIFR or colonies from local beekeepers by Indian researchers. All sampling and processing of biological material was carried out in India, and no material was exported. Publication of this study and data are exempt from prior approval under Section 4 of the Biological Diversity Act, 2002 (as amended in 2023).

All honeybee workers (10–50 individuals per colony) were collected from colonies across locations described in the metadata sheet (Supplementary Data 1). Most colonies sampled in Malaysia were wild except those *A. mellifera* samples labeled "Langstroth hive" under the hive description column. *A. cerana* samples from Langstroth hives were local bees that were temporarily housed in these hives with minimal intervention but not managed as *A. mellifera* normally are. Bees were stored in small ventilated boxes upon collection and transferred to absolute ethanol within a day. Bees were then stored at −20 °C until dissection and DNA isolation, sufficiently preserving the sample for sequencing the microbiome[68]. Our sampling strategy prioritized covering many colonies across locations to capture the diversity of each host species rather than many individuals from the same colony, which was done before[37,39].

### Dissection and DNA extraction
To decrease the ratio of DNA from the host cells to bacterial cells, we only used the hindgut which contains most of the bacterial biomass. To do so, we carefully opened the cuticle, removed the stinger and venom glands, and, finally, cut the tissue near the pylorus, below the

malpighian tubules removing the midgut and crop. Then, the hindgut tissue was transferred to a bead-beating tube containing 478 μL of sterile PBS and 2.0 mm sterile glass beads and subjected to bead-beating for 30 s at 6 m/s in FastPrep-24 5 G (MP Biomedicals). Next, 0.1 mm zirconia beads along with 250 μL of 2x CTAB, 2 μL of β-mercaptoethanol and 20 μL of proteinase K (20 mg/mL) was added and bead-beating was carried out for 30 s at 6 m/s in FastPrep-24 5 G (MP Biomedicals). The homogenate was incubated for 1 h at 56 °C. Next, 750 μL of Phenol-chloroform isoamyl alcohol (25:24:1, v/v) at pH 8.0 was added and mixed thoroughly, followed by centrifugation for 10 min at $16,000 \times g$. The upper aqueous layer was transferred to a new tube, and 500 μL of chloroform was added and mixed thoroughly. The mixture was centrifuged for 10 min at $16,000 \times g$, and the 300 μL of the upper phase was mixed with 900 μL of chilled absolute ethanol and incubated at −20 °C overnight. Next, the precipitated DNA was pelleted by centrifugation for 30 min at $16,000 \times g$ at 4 °C. After two washes with 70% ethanol, it was resuspended in 50 μL of nuclease-free water. Next, it was treated with RnaseA and purified using magnetic beads (Clean NGS). DNA concentration was measured using Qubit™ (Thermo Fischer Scientific). Bacterial absolute abundance was measured using qPCR with primers targeting the 16S rRNA region as described earlier[69] with 0.1x dilution of gut homogenate.

## Metagenomic sequencing

Library preparation was carried out using the Nextera XT library kit (Illumina), according to manufacturer instructions, with the number of PCR cycles chosen to account for the amount of DNA in samples. Illumina sequencing was carried out using Novaseq ($2 \times 150$ bp) with a target of 10–50 million microbial reads. Since there was a significant difference between host species in the total gut microbial content as determined by qPCR using universal primers, first, a run of sequencing was carried out without normalization of the amount of samples added to the library to estimate the percentage of host DNA and microbial DNA in the library made from each sample. This was useful in ensuring that there was enough DNA from the bacterial community in all the samples, including those where the library might have been dominated by host DNA due to lower bacterial biomass. Further sequencing was then carried out after pool correction, where the library was adjusted in such a way when possible that at least 15 million microbial reads were obtained (based on expectation from the first shallow sequencing), even for the samples with the lowest percentage of microbial DNA. All library preparation and sequencing was done at the Lausanne Genomic Technologies facility (GTF), UNIL. Details about all further bioinformatic analysis, including the code used, can be found on GitHub [https://github.com/Aiswarya-prasad/honeybee-cross-species-metagenomics] and Zenodo [https://zenodo.org/doi/10.5281/zenodo.13732977] in the repositories.

## Mapping to databases

To assess the amount of bacterial, host, and other reads in each sample, reads were mapped to host and MAG databases. The host database comprised draft or complete genomes of all the four host species (GCA_000184785.2, GCA_000469605.1, GCA_001442555.1, GCF_003254395.2) except *Apis andreniformis* since no genome was available in public databases for this host species. A database was compiled using representative MAGs identified for each 95% ANI cluster or bacterial species with the highest quality score (calculated as described by dRep[70] v3.4.3 using various MAG characteristics, including CheckM v1.2.2 contamination and completeness). Trimmed and quality-filtered reads were mapped to the host database using BWA-mem[71] v0.7.17 followed by the MAG database (and vice versa to account for non-specific mapping). Three samples were removed from the downstream analysis as their metagenomic reads did not map the host that they were labeled as (Supplementary Data 3) and might have faced cross-contamination during sample processing. Samples for which relatively

few reads mapped to the MAG database often were also low in bacterial biomass as measured by qPCR (Supplementary Fig. S16). Indicating that the sequencing and workflow to detect microbial diversity in each sample was otherwise robust. Initial taxonomic profiling of the reads using mOTUs[72] v3.1.0 detected the expected core marker genes from various honeybee gut microbiota members.

## Assembly and binning of MAGs

Raw reads were trimmed and quality-filtered before assembly using Trimmomatic[73] v0.39 and bbtools v39.01. We did not carry out host filtering at this stage as we have no host genome for *A. andreniformis* and did not want to bias our samples due to non-specific mapping. For some samples that had a large number of reads (19/200 samples) due to which assemblies were not successful, bbnorm v39.01 was used to down-sample regions of high depth. For samples with too many host reads due to which assemblies failed (3/200 samples), host-filtered reads were used. The detailed workflow used for assembly and binning into MAGs is described in GitHub [https://github.com/Aiswarya-prasad/honeybee-cross-species-metagenomics]. Briefly, contigs were assembled using metaSpades[74] v3.15.3. Quality-filtered reads were mapped against assembled scaffolds (all against all samples), and then scaffolds were binned into MAGs using coverage information as implemented in MetaBAT2[75] v2.12.1. The quality and completeness of the resulting MAGs were assessed using CheckM[76] v1.2.2 and low (< 50% completeness or >5% contamination), medium (50-90% completeness and <5% contamination), high (> 90% completeness and <5% contamination) MAGs were then clustered into groups averaging 95% average nucleotide identity (ANI) using dRep[70] v3.4.3. To further ensure that our MAGs represent true species and are not assembly and binning artifacts, we inferred phylogenies, including isolate genomes along with our MAGs. The rationale is that if MAGs are artifacts, they would rarely cluster together in the phylogeny and would instead be placed apart from isolate genomes. This new analysis shows that our MAGs form discrete phylogenetic clades, which include MAGs assembled from several different samples. Further, these clades often include closely related isolate genomes in cases where several strains were included from the same species. MAGs belonging to clusters without isolate genomes represent novel species not isolated before (Supplementary Fig. S4). We are, therefore, confident that the MAGs represent true species and can be used to infer meaningful phylogenies. Other studies in the field, which have used similar approaches to ours, are similarly able to establish meaningful phylogenies from MAGs[7].

Taxonomic annotation was carried out using GTDB-tk[77] v2.3.2. No manual curation of MAGs was performed; however, Kaiju[78] (nr database) v1.9.2 and Kraken2[79] v2.1.3 (custom database including bacteria, viruses, and honeybee) classification of scaffolds screened by random sampling further confirmed that scaffolds from the same MAG matched each other and the taxonomic assignment of the MAG. Each cluster of MAGs was named to reflect the GTDB species name or, in cases where none was assigned, a unique identifier, including the genus name was used. Combined with the inspection of ANI heatmaps (Supplementary Fig. S3), a final set of species names was assigned to each of the 95% ANI clusters.

## Taxonomic profiling

InStrain[43] v1.8.0 was used according to its recommended workflow for taxonomic profiling at the bacterial species and strain level. Trimmed and quality-filtered reads were competitively mapped to the database of representative MAGs using the tool bowtie2[80] v2.5.1, which presents mapping scores in a format compatible with InStrain. Among many approaches to detecting microbial species in a dataset, this stands out in its appropriate filters applied to mapped reads, including read ANI, which helps differentiate genomes of closely related species (< 95% ANI) and breadth of the genome mapped to avoid falsely identifying

species as present due to high coverage of a region shared with others or horizontally transferred. Moreover, genome coverage of a given species can be misleading when comparing communities of varying diversity, as an equally abundant species would have higher coverage in a community with low diversity. A species was considered present if at least 50% of its genome (comprising all scaffolds) was covered by the reads. The justification for the choice of this threshold is exemplified for several Snodgrassella species in Supplementary Fig. S5: briefly: there is a clear gap between samples where a given species is detected (in large green circles) or not (in small red circles). We inspected such plots for all the species represented in the database and visually verified that the breath coverage we used (50%) was valid across species. These plots are included with the dataset found in the Zenodo repository [https://zenodo.org/doi/10.5281/zenodo.13732977].

Relative coverage, when used, was obtained by dividing the coverage of each species by that of the sum of all, and qPCR copy numbers of the bacterial 16S rRNA region were used to estimate absolute abundance for comparison between samples where mentioned. For samples from India ($n = 50$), qPCR values were interpolated based on the median of the host species. The underlying data (presence-absence, relative abundance etc.) and normalization approach are explicitly indicated for each downstream analysis section. After filtering samples as mentioned earlier and removing samples with fewer than 1 million mapping quality filtered reads mapping to the representative database of MAGs, a total of 195 samples were included in the analysis (*A. mellifera* 34, *A. cerana* 45, *A. dorsata* 44, *A. florea* 42 and *A. andreniformis* 30).

## Host specificity and strain-level profiles

Host specificity was calculated for each bacterial species as the Rohde's index of specificity[42] using the prevalence of the species across hosts. Strain-level composition profiles for each bacterial species represented in the MAG database were generated per sample using InStrain in database mode. The percentage of variant sites was calculated as the percentage ratio of the number of SNVs to the total length of the MAG covered. A site is counted as an SNV if the site is covered at least 5x and the allele is present at a frequency of at least 0.05. Regions mapping with a read ANI of <0.92 (might be cross-mapping of reads from a different species) or coverage of <1x were not considered. To compare the strain profile of samples, the popANI measure, as comprehensively described by InStrain[43], was used. Briefly, a value from 0 to 1 was calculated for each species for each pair of samples in which the genome was covered at least by 5x and at least half of the genome was covered in both samples. A position of the genome was considered if there was more than one allele with a frequency of at least 0.05. If both samples share no alleles in that position, it is counted as a popANI substitution. The value for that pair of samples was then calculated as the ratio of positions containing popANI substitutions and the number of positions compared. This value was used to create a matrix of Jaccard distances between all pairs of samples. Finally, each of the bacterial species was either considered host-specific, if they were detected in only one host species, or shared if not. Shared species were considered host-specific at the strain level if samples showed significant clustering by host species ($p < 0.01$) according to a PERMANOVA test.

## Phylogeny construction and co-diversification test

To gain a rough estimate of the taxonomic level at which to expect co-diversification, we computed a core genome phylogeny using the nucleotide sequence of 120 bacterial marker genes for all medium and high-quality MAGs identifying core genes using GTDB-tk, aligning in a codon-aware manner using MACSE[81] v2.07 and constructed using IQ-TREE[82] v2.2.2.7. From this tree, the median substitutions per site were 0.014 for MAGs of the same species, 0.39 for different species of the same genus, and 2.65 for different genera (Supplementary Data 7).

Bacteria evolve at a rate in the order of magnitude of $10^{-3}$ substitutions per site per million years[83]. Since the *Apis* clade is about 30 million years old[84], gut microbes that diverged during this timeline could be as divergent as different species within a genus or closer, confirming that genus-level trees are appropriate for these tests.

OrthoFinder[85] v2.5.5 was used to infer orthologous gene families of all medium and high-quality MAGs for each genus along with previously published genomes of the same genus from other environments as outgroup (details of genomes used in metadata files included in the Github [https://github.com/Aiswarya-prasad/honeybee-cross-species-metagenomics] and Zenodo [https://zenodo.org/doi/10.5281/zenodo.13732977] repository). Phylogenies were constructed using IQ-TREE using the amino acid sequences of genus-level single-copy core genes aligned by MAFFT[86] v7.520. All medium and high-quality MAGs (including several per bacterial species) were included in the phylogenies in order to preserve maximum information and power of resolution. Since medium-quality MAGs were included in phylogenies, core genes can be randomly missing in some MAGs. Naively defining core genes as those present in all MAGs would result in the exclusion of most orthologs. Hence, as implemented in OrthoFinder v2.5.5, when <1000 single-copy orthologs are present in all MAGs, it iteratively identifies the orthologs that are found once in N-1 MAGs until either considering one less MAG does not double the number of orthologs or more than half the MAGs are dropped. Core single-copy orthologs identified were in agreement with those orthologs identified as core by mOTUpan[87], which uses a Bayesian approach informed by the completeness score of MAGs. The number of orthologous gene families (OG) used for core genome phylogenies and the minimum percentage of MAGs of the genus in which each OG was detected (e.g., 100% means all OGs were detected in all MAGs and 50% means every OG was found in 50% or more of the MAGs): Apibacter—1096 (94.5%), Apilactobacillus—704 (81.8%), Bartonella—705 (81.5%), Bifidobacterium—118 (84.0%), Bombilactobacillus—194 (89.4%), CALYQJ01—883 (89.5%), CALYQQ01—1221 (84.6%), Commensalibacter—1120 (96.6%), Dysgonomonas—223 (86.5%), Entomomonas—1144 (91.7%), Frischella—984 (85.9%), Gilliamella—189 (88.0%), Pectinatus—1143 (97.3%), Pluralibacter—1419 (100.0%), Saezia—1000 (92.3%), Snodgrassella—799 (91.5%), WRHT01—1897 (100.0%). Further information about the parameters and best-fit model chosen to compute the phylogenies can be found in the log files included in the corresponding directory containing IQ-TREE results in the Zenodo [https://zenodo.org/doi/10.5281/zenodo.13732977] repository.

A host phylogeny inferred based on complete mtDNA sequences of honeybee species rooted by a bumble species published before[88] was used for co-diversification tests. Tests were carried out using the Hommola cospeciation test[89] (implemented in Python) on all nodes containing at least 7 MAGs from at least three host species and with a maximum tip-tip distance not exceeding 1 for 100 permutations. This node-by-node comparison was then carried out on trees with tip labels artificially permuted randomly across ($N = 100$) second-order permutation test to estimate the number of nodes displaying significant results by chance, as described in an earlier study[45]. See Supplementary Fig. S20 for a detailed schematic of this approach. Thresholds were set at various levels for comparison of the number of significant nodes in the original tree and the median number of significant nodes in tip label permuted trees: relaxed ($p < 0.05$), medium ($p < 0.05$ and $r > 0.75$), and strict ($p < 0.01$ and $r > 0.75$). Finally, if nodes including outgroups were picked up as significant (2 out of 15), they were excluded because the outgroup genomes come from sources other than the honeybee species being compared. The remaining significant subtrees were visually inspected for congruence between symbionts and their hosts. Specifically, we examined whether the branch lengths separating symbionts from different host species were consistent with the estimated divergence times of their hosts. For instance, given that all host species diverged several million years ago, we expected

symbionts associated with different hosts to exhibit clear signs of sequence divergence in our phylogenetic trees.

## Gene functional annotation and profiling

Open reading frames (ORFs) were identified on all assembled scaffolds of each sample using Prodigal[90] v2.6.3. ORFs marked partial or shorter than 300 bp were removed. Further, those ORFs detected on scaffolds identified as being of eukaryotic origin by Whokaryote[91] v1.1.2 or unclassified as bacteria by Kaiju (nr database) were also excluded. The remaining ORFs were then annotated using DRAM[92] v1.5.0 to obtain KEGG ortholog (KO) identities. Finally, the gene catalog comprising all these ORFs was assigned to CAZyme families as implemented in Cayman[93] v0.9.2.

Trimmed and quality-filtered reads from each sample were mapped back to their respective assemblies using bowtie2. Low-quality (Q < 20) alignments were removed, and coverage per gene/ORF was calculated using BEDtools[94] v2.31.0. Genes were considered detected if >5 reads were mapped to it and >90% of the bases in the gene were covered by at least a read. The abundance of each gene was then calculated in terms of reads per kilobase per million (RPKM) calculated as follows:

$$RPKM = \frac{\#Mapped\ reads}{\left(\frac{Gene\ length}{1000}\right) * \left(\frac{\#Total\ reads}{10^6}\right)}$$

For KOs and CAZymes, summed RPKMs of constituent genes were used. To estimate the average count of CAZymes of interest (all families other than GT and AA families) in each genus per individual, the RPKM of all CAZymes in each genus was summed per sample, and then the average of this sum across samples was converted to a relative proportion to represent the relative contribution of each genus on average in each host species and plotted. Only genera with a prevalence >0.05 in the respective host were considered.

To infer biological pathways represented within each MAG, we collected the KOs detected within each MAG and used MinPath[95] v1.4, a parsimony approach which aims to explain the set of protein (KO) families using the minimum number of pathways. Further, we used MaAsLin 2[96] v1.16.0, which was developed to assess multivariable associations of microbial community features with complex metadata. With the presence-absence matrix of pathways as the dataset and 'genus' as the explanatory variable we identified the most likely functional similarities driving certain genera to cluster together in the PCoA plot of all KOs per MAG (Fig. 4C).

## Statistical tests and bioinformatic analyses

All details about the specific parameters used for each tool and their versions, along with conda v23.3.1 environment specifications, are included in the Github [https://github.com/Aiswarya-prasad/honeybee-cross-species-metagenomics] and Zenodo [https://zenodo.org/doi/10.5281/zenodo.13732977] repositories. In general, bioinformatic workflows were implemented using Snakemake v7.28.1 to parallelize jobs on a computing cluster at University of Lausanne, preprocessing and filtering using Python scripts, visualization using ggplot with several related packages in R v4.3.1 and iTOL v6 for phylogenetic trees and arrangement and decoration of plots using Adobe Illustrator. Unless otherwise specified, statistical tests of the results of each section were carried out as follows. Significance in boxplots was tested using the Wilcoxon Rank Sum test (significant if $p < 0.05$). Distance matrices were calculated using the method indicated in their respective plots. The package Betapart[97] v1.5.6 was used to estimate the turnover component of the Jaccard and Sorensen distances. Multivariate analyses were visualized using PcoA or NMDS and tested using the adonis2 function implemented in the R package Vegan v2.6-4. The effect size was adjusted for sample size to get $\Omega^2$ using the adonis_OmegaSq function from the package Russel88/MicEco v.0.9.15 [https://zenodo.org/badge/latestdoi/83547545]. To test for phylosymbiosis, we minimized bias due to pseudo-replication by performing 1000 permutations. Each permutation included five randomly selected samples, one of each host species, and tested for correlation between bacterial and host distances using a Mantel test (as implemented in the Vegan package). The p-values were adjusted using the FDR method.

## Reporting summary

Further information on research design is available in the Nature Portfolio Reporting Summary linked to this article.

## Data availability

Raw metagenomic data and MAGs have been deposited to the NCBI Sequence Read Archive (SRA) under the Project ID PRJNA1157353. The collection of MAGs, some important intermediate files, and tables can be found in the Zenodo repository [https://zenodo.org/doi/10.5281/zenodo.13732977]. The phylogenetic trees are available to visualize and download on iTOL [https://itol.embl.de/shared/Aiswarya].

## Code availability

Code and details of parameters and software used are available[98] on GitHub [https://github.com/Aiswarya-prasad/honeybee-cross-species-metagenomics] and archived at the Zenodo repository [https://zenodo.org/doi/10.5281/zenodo.13732977].

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

## Acknowledgements

We thank the following people and organizations for supporting honeybee sampling and access. The Nature Inspired team (John Chan, Steven Wong, & JC Tan), Mr. HoneyBee Farm, The Orang Asli community (Temengor, Perak), Sherwani Honey Bee, Mr. Wee, Mr. Steve, and MY Bee Saviour. The Cayman team (Quinten Ducarmon, Nicolai Karcher, and Christian Schudoma) for their support with early access to their software and custom annotation of our dataset. The Lausanne Genomics Technology Facility (GTF) team at the University of Lausanne. This work was supported by an SNSF Spirit grant (grant number 189496)—P.E. and S.H.Y., an SNSF Consolidator grant ('GLOBEE,' grant number 213860)—P.E., and the NCCR Microbiomes, a Swiss National Centre of Competence in Research (grant numbers 180575 and 225148)—P.E., funded by the Swiss National Science Foundation. We thank Charlotte Huyghe and Malick Ndaye for their feedback on the revised manuscript.

## Author contributions

Conceptualization: P.E., S.H.Y., A.P., and F.M.; Methodology: A.P., F.M., and P.E.; Software: A.P.; Validation: A.P.; Formal analysis: A.P.; Investigation: A.D.P., A.P., R.S., and A.S.; Resources: P.E., S.H.Y., and A.B.; Data curation: A.P.; Writing—original draft: A.P.; Review and editing: P.E., F.M., A.P., A.B., and S.H.Y.; Visualization: A.P.; Supervision: P.E., S.H.Y., F.M., and A.B.; Project administration: P.E., S.H.Y., and A.B.; Funding acquisition: P.E. and S.H.Y.

## Competing interests

The authors declare no competing interests.
