## [Transparent Peer Review file · Nature Communications]

Evolution of gut microbiota across honeybee species revealed by comparative metagenomics

Corresponding Author: Professor Philipp Engel

Version 0:

Reviewer comments:

Reviewer #2

(Remarks to the Author)

Prasad and co-authors performed a metagenomic analysis of honey bees from five species. After removing host reads, they assembled contigs of the microbial metagenomes and annotated the gene functions. They identified unique and common microbes among the bee species and strain level variance. The authors did not find host-symbiont co-diversification, which contradicts previous papers where microbes of *A. cerana* and *A. mellifera* show diversification. In principle, this paper describes the microbial communication among bee species. Previous publications from the same group considered the impact of the season (Ellegaard and Engel, 2019, NC) or the variance between two bee species (Ellegaard et al. 2020, CB). However, in this paper, the sample collection is at a single time point, which has no power to illustrate any symbiont gain, loss, or switch. Additionally, the chance of false positives to infer bacterial species could be high when using metagenome sequencing. Species-specific PCR or other validation is required to ensure the inferred species is correct. Additionally, I tried to run the pipeline, but the raw data is not accessible because the provided link (<https://zenodo.org/doi/10.5281/zenodo.13732977>) for data deposit is invalid.

(Remarks on code availability)

The raw data is inaccessible because the provided link (<https://zenodo.org/doi/10.5281/zenodo.13732977>) for data deposit is invalid. Thus, it is not possible to validate the code.

Reviewer #3

(Remarks to the Author)

The authors have conducted a very interesting study characterizing the microbiome of five *Apis* species: *A. cerana*, *A. dorsata*, *A. florea*, *A. andreniformis*, and the well-known *A. mellifera*, which serves as a key model system for microbiome research. While the microbiomes of *A. mellifera* and *A. cerana* have been previously characterized by the same research group also using shotgun sequencing (L91), studies on the remaining three species are sparse, likely due to their status as non-managed species. The authors analyzed a total of 195 samples (from 40 individual hives) and identified 150 dereplicated metagenome-assembled genomes (MAGs) of medium to high quality. Community and functional analyses were conducted, and by leveraging the phylogenetic relatedness of these species within the monophyletic Apini clade, the study also addresses a critical question in microbiome research: the potential for codiversification of microbiomes with their hosts. The authors state (L102) that "...ancient gut microbiota-host interactions can be maintained in the absence of co-diversification", which I comment below. The figures are thoughtfully designed to highlight the main findings, and the manuscript is clear and well-written, although a few additional details in the introduction would be helpful. The methods and software used were appropriately applied, with well-documented code made available.

The novelty of this study is the in-depth characterization of three non-model *Apis* species using shotgun metagenomics, providing greater resolution to investigate the evolution of the honey bee microbiome. Additionally, this is the first study to test for co-diversification in honeybees without relying on 16S rRNA data. These findings can certainly contribute to the broader microbiome field, which so far has observed co-diversification patterns in other systems such as the microbiome of primates and humans. In this study, however, the authors suggest that the gains and losses of bacterial species appear to be among the most significant drivers of honeybees' microbiome composition, a pattern also observed and suggested in previous studies on corbiculate bees (*Apidae* clade) and stingless bees (*Meliponini* clade) (doi: 10.1126/sciadv.1600513, 10.1128/mbio.02317-21).

Major comments

My only concern relates to the main conclusions of the manuscript stating that symbiont loss and gain—rather than co-diversification—shapes honeybee gut microbiota diversity and function. The authors used genus-level phylogenies to test for co-diversification (L200) because the divergence times within genera are roughly equivalent to those in the *Apis* genus (Apini tribe). However, this implies that diverged bacterial species and strains could exist within the Apini tribe. Thus, a robust test for co-diversification would necessitate a well-resolved symbiont phylogeny, particularly at the strain level. Ideally, collecting strain isolates from prevalent genera across different bee species and conducting transplant experiments would enhance the analysis. Nonetheless, given the challenges of working with non-model honeybee species, metagenomics remains a gold standard approach. With that in mind, would it not provide better resolution (although still not the ideal scenario) to dereplicate at 99% average nucleotide identity (ANI) and perform co-diversification tests at a finer scale? Discussion on dereplication in reference 10.1128/mSphere.00971-19.

As mentioned in the manuscript, previous studies in other model systems suggest that only a small percentage of clades (taxa) within a microbial community would codiversify with their host. Nevertheless, this is an important finding as it highlights an evolutionary mechanism driving microbiome evolution, and thus deserves further exploration. Looking at Table S6 (L215), it appears that significant codiversifying nodes were identified at the various thresholds used. If I have misinterpreted this, could you please clarify the table? Including the expected codiversification pattern by chance might be helpful, as I was unable to find this comparison in the tables or figures. If only a single bacterial clade appears to be codiversifying, it may be valuable to use the most complete MAGs to test for gene functions that are significantly overrepresented in this codiversifying clade compared to non-codiversifying clades, and to calculate gene dN/dS ratios, for example.

Minor comments

- L83: I mostly recall studies suggesting gain and loss of microbial associations and discussing the potential for co-diversification. Add references for “Co-diversification has been suggested as the primary mode of evolution” to support this statement.
- L96: It would be helpful for the reader if the authors could better contextualize the host species, including information on where they occur and the main differences in their biology. This additional context would also enhance the discussion later in the manuscript, such as in L301.
- L110: Were all samples collected from wild nests? Make sure this detail is also included in the methods section.
- L113 and Fig S2: Double-check number “1959” or “1958” MAGs.
- Fig S8: It appears that some bacterial species are represented with the same color, making them difficult to distinguish.
- L164: You introduce the term 'phylosymbiosis' here, which was not mentioned in the introduction. It can be helpful to clearly define and differentiate this term from co-diversification and co-evolution in the introduction. Distinctions are discussed in studies like doi: 10.1111/jeb.14221.
- Fig 2B and C: Reading the methods, it was not entirely clear whether the input data for host-specific analysis was at the strain level and how it was used here. Does inStrain provide information on whether specific MAGs with particular SNPs are present in a sample?
- L230: Figure “Fig. 4A”.
- L235: Make sure to cite Fig. 4B here. In fact, the entire paragraph could include citations to Fig. 4 wherever relevant data is discussed.
- L303: Clarify what you mean by “specific ecological niches” and add a reference. Provide more details on whether this relates to diet, nesting locations, or other factors.
- L342: Italicize *A. dorsata*.

(Remarks on code availability)

Reviewer #4

(Remarks to the Author)

Prasad et al 2024 – Reviewer comments

The authors performed shotgun sequencing on 5 honey bee species hind guts. They generated a dataset to show which bacterial species were present in which honey bee species and compared diversity across hosts. The authors noted the presence of specialist vs generalist bacterial species. They also found in some cases, for shared species, host-specificity exists at the strain level. The authors constructed core genome phylogenies. They found novel clades of bacteria, many in *Apis dorsata*.

The authors also tested for codiversification (Hommola cospeciation test) in 9 genera. They conclude codiversification did not occur in the majority of species/host pairs. In phylogenies, host specific clades are interspersed by generalist clades. The authors also looked at functional differences in the microbiota across the bee species. They looked at KO functional groups of the microbiota and found them to be distinct across all host species. Some host specific genera (eg, *Dysgonomonas*) had their own clusters, suggesting unique function within a host. A few genera (*Saezia* and *Entomomonas*) clustered with unrelated genera (eg, *Snodgrassella*), suggesting functional replacement. In the case of *Dysgonomonas*, the authors provide evidence that this genera helps to degrade plan fiber

Overall, the authors have performed a comprehensive study on bee microbiome diversity in different honey bee species. The experimentation and initial analysis appear to be sound. However, the interpretation of the analysis and cutoffs used seem to

be leading to the opposite conclusion that I draw when I look at these data (ie, that co-diversification is occurring, with potentially other evolutionary drivers contributing). See comments below.

This work has the potential to be impactful for evolution and host specificity of bee microbiome members, provided the conclusions are modified. Specifically, the data show co-diversification occurs, but is not the sole driver of evolution in bee microbiome members (as evidenced by weak-moderate correlations for some nodes).

Major comments

1. The claim that “honeybees and their gut bacteria did not codiversify” (line 217-218) is too strong. The data (tables S5 and S6) show a large number of the bacteria test have significant p values. The authors use a cutoff of $r > .75$ (as in Sanders et al 2024), but this omits significant correlations that are still weak/moderate/and at the lower end of strong (<https://sphweb.bumc.bu.edu/otlt/MPH-Modules/PH717-QuantCore/PH717-Module9-Correlation-Regression/PH717-Module9-Correlation-Regression4.html>). For example, 52/103 nodes with significance (table s5) have an r of .4 or greater, indicating a moderate-strong correlation. Using this lower r, one would conclude that co-diversification is present in over half of the relationships tested, at a moderate amount. If an R value of $> .2$ (weak correlation) is used, then 88/103 nodes display co-diversification, with weak correlation.

I believe a more appropriate conclusion would be that since most nodes have weak or moderate correlations, the data suggest co-diversification is occurring, but is not the only process driving evolution. Ie, another process like host-switching (as alluded to) could play a larger role in driving evolution in nodes with low R values.

I would recommend to flesh this section of the results out by including some tanglegrams and describe the bins of nodes in strong, moderate, and weak correlation, as done above.

2. Related to comment #1, the language throughout, including the title, should be softened to convey that other forces, in addition to co-diversification, shape evolution of bee microbes

3. The language on host switching as a driver for evolution is also too strong (Eg, line 195-196, line 324). Host switching can be proposed as a possible additional factor driving evolution.

4. Figure 4B shows distinct clusters of bacterial KO annotation profiles based on host species. Ie, bacteria cluster together based on host despite being in different clades. This seems to also provide evidence for co-diversification between host and microbiota. The authors might consider reframing how they discuss these data in the results to indicate this point.

It might be helpful to see Figure 4B with another dimension – bacterial genus (perhaps use shape host species and color for bacterial genus). It is unclear if bacteria cluster better within host species or bacterial genus. If the former, than this can be taken as evidence in favor of co-diversification.

Specific comments

1. Line 129-130: Be specific in the text that this study increases MAGs for species that have been undersurveyed to this point (specifically, *A dorsata* and *A florea*).

2. Line 226: define KO identifier

3. Line 230: Fig 2A should be Fig 4A

4. Lines 232-Lines 247: Figs 4B and 4C should be cited as they are discussed in the text/

5. Line 242-243: Can you provide any additional information about how the KO annotations in *Saezia* and *Entomomonas* are similar to *Snodgrassella*? Ie, any hypotheses on what function might they be replacing?

6. Line 258-265: There is a logic jump here. *A dorsata* contains multiple bacterial genera that contain pectin/cellulose degrading enzymes. *Dysgonomonas* also has polysaccharide degrading enzymes. It is unclear, as written, that the PULs in *Dysgonomonas* are expected to degrade unique polysaccharides. If there is literature suggesting PULs have a range of substrate specificity and presence of a unique PUL suggests a unique polysaccharide substrate, then this should be cited. The phrase “expands the functional capability to degrade plant-derived fibers” needs more explanation.

(Remarks on code availability)

n/a

Version 1:

Reviewer comments:

Reviewer #2

(Remarks to the Author)

The authors partially addressed my concerns. In principle, this study collected bees of five species and sequenced them individually with Illumina paired reads. Then, the authors analyzed the microbial community with these reads. Several studies have confirmed bee species-specific microbiota and distinct microbes either at the species level or the strain level, so it is not surprising to find community variance among the five bee species. However, including the giant and dwarf honeybees improves our understanding of bee symbiosis.

If the authors conclude that a bee species gains a symbiont, it means this bee species didn't have this symbiont before, but has it now. It is the same for symbiont loss. There is no time series analyses for each species in this study. Thus, it is a single time point data. When referring to the host switch, who is the native host? Overall, robust evidence is needed to support this very important conclusion to make sure it is correct. The data didn't include any ancient symbionts, how can you determine that ancient microbiota-host interaction can be maintained in the absence of strict co-diversification? The data heavily depends on the available genome assemblies when assigning the contigs to the microbial species. Symbionts in *A. mellifera* and *A. cerana* are well represented. While, giant and dwarf bees are underrepresented. The authors performed several analyses to reduce the bias, which is good. However, bias will still exist. The result interpretation must be careful, that is why I asked for species-specific PCR validation. I shared this with other reviewers and am concerned about the conclusion that there is no co-diversification. Looking at Figure 3, the symbiont of the same bee species scattered among clusters, which is extremely weird. Previous papers showed a clear symbiont diversification among bee species (for example, Kwong et al. *Sci. Adv.* 2017). The current phylogenetic tree suggests that the variance is higher (or equal) within a bee species than among bee species, which is unlikely to be true. I suggest the authors use all the contigs of microbiota as a reference to build a phylogenetic tree, and compare this tree with the host phylogeny. I also suggest the authors confine the data interruption within symbiont community variance, avoid over stating the results.

(Remarks on code availability)

Reviewer #3

(Remarks to the Author)

I appreciate the author's response to my main concern (reply 3.2). It is now clear that all MAGs, including strains, were used for the analysis rather than only dereplicated MAGs at the species level—this approach makes more sense.

Thank you also for clarifying the results in Table S6 (reply 3.3) and updating it for better clarity. The addition of tanglegrams to visualize nodes that pass various thresholds (Fig. S12) is a valuable supplement, as it more clearly illustrates that "several nodes show congruence but only cover three different host species, with one of them represented by just one or two MAGs while others have several, which can lead to increased R and significant p-values". I also appreciate the revision to the text, adding a note comparing these findings to observations in hominid microbes.

The authors have also addressed all my minor suggestions.

(Remarks on code availability)

Reviewer #4

(Remarks to the Author)

In this study, Prasad et al performed shotgun sequencing on the hindgut of multiple bee species. Among other analyses, the authors tested for co-diversification. Due to the high R value cutoff they imposed, they conclude co-diversification did not occur in most species pairs. As part of this analysis, the authors omit significant correlations below .75 (eg, moderate-strong correlation of $.4 < R$ and weak correlation of $.2 < R$). Because of this omission, the authors conclude most nodes do not correspond with co-diversification.

In my previous comments, I noted that inclusion of nodes with an $R < .4$ or $< .2$ would demonstrate that co-diversification is occurring, but is not the sole process driving evolution (ie, host switching may also occur).

I recommended:

1. Fleshing out the visualization of the analysis assessing co-diversification
 - Include tanglegrams
 - Visualize strong, moderate, and weak correlation
2. Change the conclusion (including the title of the paper) to reflect that co-diversification occurs, but is only a partial driver of evolution.

The authors have addressed point 1, but not point 2.

The authors have added tanglegrams as part of Fig S12. In line 232 of the main text, the authors note "Closer inspection of the subtrees from these nodes showing the highest signals of potential co-diversification for each genus showed little congruence with host phylogeny". However, again, I reach the opposite conclusion looking at these data. Each of the tanglegrams shown have a high R value, indicating a strong correlation between host and symbiont. Further, most of the tanglegrams shown visually depict a high level of concordance between host and symbiont, with very little "tangling" (Frischella has some tangling, but also some concordance). These data support the conclusion that co-diversification occurs in most host-symbiont pairs to an extent, with host-switching also likely contributing to evolution.

Fig S13 depicts tanglegrams with varying levels of R value stringency. The relaxed model, illustrates well that in nodes with

weaker correlations, both co-diversification and host switching occur.

The interpretation that host-switching and co-diversification both contribute to evolution are consistent with previous work. Koch et al 2013, for example, shows that both co-diversification and host-switching contribute to evolution in *Snodgrassella* and *Gilliamella*. In this work, no R value cutoff is used for Mantel tests. (R values were found to be .680 and .332 for these genera respectively). Tanglegrams from this study also illustrate a mix of concordance and “tangledness”, similar to in Prasad et al 2024. Koch et al conclude these data, “reject a strict cospeciation of *Snodgrassella* and *Gilliamella* with their bee hosts. The association between these bacteria and bees appears to allow for a considerable degree of host switching... Nevertheless, for both *Snodgrassella* and *Gilliamella*, we found a significant correlation between the 16S rRNA gene bacterial phylogeny and the host phylogeny. The reconciliation analyses furthermore suggest the possibility for a number of cospeciation events.” In other words, the range of strength of correlations they observe point to both co-diversification and host-switching as drivers for evolution. Prasad et al 2024’s data align with those of Koch et al 2013, so the conclusions should be similar.

With this in mind, in their revised manuscript, the authors still do not acknowledge the role that co-diversification plays (in conjunction with host-switching) in evolution.

In order to support publication of this manuscript, I recommend:

1. The authors update their conclusion to reflect that both co-diversification and another process, such as host-switching, contribute to species evolution.
2. Update their title to reflect this conclusion.

(Remarks on code availability)

Version 2:

Reviewer comments:

Reviewer #2

(Remarks to the Author)

A single time point collected samples can't infer symbiont gain or loss. I suggested the authors confine their conclusion to symbiont community dynamics among bee species, which is also interesting. However, the authors insist on drawing this essential conclusion on the evolution of bee symbiont gain and loss, which is substantially overstated from their data

(Remarks on code availability)

Reviewer #4

(Remarks to the Author)

In my previous review, I recommended that the authors update their conclusion and title to reflect that both co-diversification and another process, such as host-switching, contribute to species evolution. Despite some minor changes in wording, the authors still conclude that co-diversification occurs “not more than expected by chance” and that host-switching predominates.

The authors highlight that the partial lack of concordance between *Frischella* and host phylogeny exemplifies that host-switching predominates in the bee gut microbiome. However, it appears that these data show that the evidence for co-diversification in *Frischella* is simply not as strong as in the other bacterial clades. Furthermore, the *Frischella* example seems to be the exception to the rest of the data – all of the other bacterial clades in the Fig S12 tanglegrams (*Bartonella*, *Bifidobacterium*, *Bombilactobacillus*, *Dysgonomonas*, *Commensalibacter*, *Lactobacillus*, *Snodgrassella*) show evidence for co-diversification, based on congruence between host and bacterial branches. The data presented, therefore, do not seem to support the conclusion that host-switching predominates over co-diversification. In fact, the data (Fig S12, tanglegrams) lead to the opposite conclusion in my view -- that co-diversification is common and host-switching is the rarity.

(Remarks on code availability)

Version 3:

Reviewer comments:

Reviewer #3

(Remarks to the Author)

I appreciate the authors' efforts to address reviewer concerns. The revised title and removal of earlier overstated claims

regarding the predominance of host switching result in a more balanced interpretation of their findings. Overall, the manuscript (as well as the responses to the reviewers) is technically sound and includes appropriate analyses. The conclusions are now more cautiously framed, as in L116: "the evolution of specialized gut microbiota-host interactions in honeybees has been shaped by symbiont loss and gain, and co-diversification." However, the primary contribution of this conclusion remains largely confirmatory of previously described patterns in bee microbiome evolution.

Limited novelty: While the analyses are carefully conducted and the recovery of many novel *A. dorsata* MAGs is valuable, the study's main conclusions largely reinforce what has already been proposed. As the authors themselves acknowledge: "Similar observations have been made for prominent gut microbiota members in stingless bees (Cerqueira et al. ISME 2021) and similar processes have already been proposed to also shape the microbiota of honey bees (Kwong et al. Sci Adv)." Their central finding (L369) that "our findings point to an important role for symbiont gain and loss, as well as host-switching, in shaping these specialized gut communities" essentially confirms established dynamics.

Co-diversification Analysis: Although I must note that host–microbiome co-evolution using such methods is not my specific area of expertise, the analyses and explanations provided by the authors seem appropriate and (mostly) carefully interpreted. However, the use of an $r > 0.75$ cutoff to define co-diversifying clades may be overly conservative — a point raised by reviewer #4, which I agree with. Many clades show moderate levels of phylogenetic congruence (52 out of 103 with $r > 0.4$, Table S5), which may indicate real co-diversification signals that are being overlooked under strict thresholds. The authors argue, for example, that "Low r -values accompanied by significant (low) p -values can arise due to a large number of MAGs." However, this explanation may be less relevant here, given that their dataset includes only 1,959 medium+high-quality MAGs, in contrast to the much larger dataset in Sanders et al. (9,460 high quality MAGs). Also, in sympatric *Apis* species (sharing environmental microbial pools) that also have significantly different life cycles than the other systems being compared, as mammals where such analyses were also done, detecting clear co-diversification signals may be inherently more difficult and other cutoffs maybe should be considered.

Interpretation: I appreciated the detailed view of co-diversifying clades in Supplementary Figure S12 and S13. It effectively illustrates cases where "there is no internal phylogenetic branch congruence between clades of host-specific symbionts and the host phylogeny, although phylogenetic clustering by host species", but also reveal cases where moderate phylogenetic congruence signals may warrant closer consideration (e.g., Figure S12B). Also, "no internal phylogenetic branch congruence" could reflect differences in evolutionary rates rather than definitively pointing to host switching, and this possibility should be more carefully acknowledged in the discussion. Additionally, while maybe this was not the main question and is not testable using the co-diversification methods applied here (which require the symbiont to be present in multiple host species), the consistent restriction of certain taxa to a single *Apis* host (e.g. *Dysgonomonas* mostly in *A. dorsata*) could reflect a long-term host association and raise questions about possible host-specific coevolution.

(Remarks on code availability)

Version 4:

Reviewer comments:

Reviewer #3

(Remarks to the Author)

I appreciate the authors' thorough revisions. They provided detailed responses and made additional minor changes throughout the manuscript. My comments have been addressed, and I have no further suggestions.

(Remarks on code availability)

Our responses are highlighted in blue in the following text. Line numbers correspond to the marked-up manuscript (with All Markup enabled and displayed inline).

REVIEWER COMMENTS

Reviewer #2 (Remarks to the Author):

Prasad and co-authors performed a metagenomic analysis of honey bees from five species. After removing host reads, they assembled contigs of the microbial metagenomes and annotated the gene functions. They identified unique and common microbes among the bee species and strain level variance. The authors did not find host-symbiont co-diversification, which contradicts previous papers where microbes of *A. cerana* and *A. mellifera* show diversification.

We appreciate the reviewer for taking the time to review our manuscript and providing feedback. We have addressed their concerns in our responses below.

In principle, this paper describes the microbial communication among bee species. Previous publications from the same group considered the impact of the season (Ellegaard and Engel, 2019, NC) or the variance between two bee species (Ellegaard et al. 2020, CB). However, in this paper, the sample collection is at a single time point, which has no power to illustrate any symbiont gain, loss, or switch.

2.1 - Reply: We respectfully disagree with the reviewer. First, it is incorrect that our bees were collected at a single time point. Second, as demonstrated by several previous studies (Moeller *et al.*, 2016; Sanders *et al.*, 2023; Rühlemann *et al.*, 2024), sampling at different time points is not needed for co-diversification testing. Thus, the reviewer's point that our study lacks the power to illustrate any symbiont gain, loss, or host switch is not justified. Please see our detailed responses below for each of our two major arguments:

1/ We sampled bees from several colonies >1000km apart and over different dates through a period of 15 months. Our approach of sampling individuals from several colonies covers intra-species variation. The sampling was deep enough to cover the diversity of bacterial species in each host species (as demonstrated by the plateauing of the #species vs #bees curves in Supplementary Fig. S6). This presents the most thorough sampling of some of these honeybee species for gut microbiome analysis to date, as earlier studies (including those mentioned by the reviewer above) have focused on sampling more individuals but from one or two nearby colonies.

2/ The goal of our paper is to describe changes in microbiota composition across an evolutionary timescale, not ecological timescale (which we believe this reviewer considers as 'microbial communication among bee species' and for which one could indeed argue that longitudinal data would be useful). Our aim was to look at the evolutionary processes that have shaped the bee gut microbiota of related host species over millions of years. These processes can include co-diversification but also symbiont gain, loss and host switches, which are inferred using a phylogenetic comparative approach by comparing related

host species, taking the intra-species variation of the microbiota into account (which we did by sampling bees from different colonies of each host species from a wide range of locations).

Additionally, the chance of false positives to infer bacterial species could be high when using metagenome sequencing. Species-specific PCR or other validation is required to ensure the inferred species is correct.

2.2 - Reply: We agree that there are two important challenges when characterizing microbial communities using MAGs from metagenomic data:
1/ Defining valid bacterial species units (i.e., not inferring artifact species or large chimeric bins due to incorrect binning of contigs)
2/ Properly assessing the distribution of bacterial species across samples (i.e. correctly inferring their presence in some samples)

We discuss each of these points in more detail below. Briefly, we have carried out additional checks and analyses to ensure the false-positive rate is low and that we can rely on our data to test our main hypothesis.

1/ Defining valid bacterial species. We have used the most up-to-date approach to define bacterial species from metagenomic data using the 95% ANI threshold (Jain *et al.*, 2018). This approach is standard and is widely used and accepted in the field (Olm *et al.*, 2021). Yet, we provide this additional explanation in the methods: Lines 524 – 534 and carried out additional analysis added as Supplementary Fig. S4 in the manuscript file and posted below.

The methods section now reads as follows: “[...] To further ensure that our MAGs represent true species and are not assembly and binning artifacts, we inferred phylogenies, including isolate genomes along with our MAGs during the revision of the paper. The rationale is that if MAGs are artifacts, they would rarely cluster together in the phylogeny and would instead be placed apart from isolate genomes. This new analysis shows that our MAGs form discrete phylogenetic clades, which include MAGs assembled from several different samples. Further, these clades often include closely related isolate genomes in cases where several strains were included from the same species. MAGs belonging to clusters without isolate genomes represent novel species not isolated before (Supplementary Fig S4). We are, therefore, confident that the MAGs represent true species and can be used to infer meaningful phylogenies. Other studies in the field, which have used similar approaches to ours, are similarly able to establish meaningful phylogenies from MAGs⁷.”

Supplementary Fig. S4. Phylogenies of MAGs and isolate genomes. Core genome phylogenies, including all high- and medium-quality MAGs of the bacterial genus and selected isolate genomes from the GTDB database for several bacterial species. Tree leaves corresponding to isolate genomes are indicated by a black circle. Bacterial species represented by clades of MAGs also harbor isolate genomes within.

2| Profiling bacterial species abundance across samples. We have now added text in the methods to clarify this section: Lines 555 – 563 and one supplementary figure (Supplementary Fig. S5 in the manuscript file and posted below)

Briefly, to properly assess the distribution of each bacterial species in each sample and minimize false positives, we use read-mapping to a database of high-scoring (based on size, completeness and contamination as estimated by dRep) representative MAGs rather than relying just on whether a MAG of that species was recovered from the sample. Within a given sample, while some reads from a related species might map to the representative MAG of a focal species, unless the focal species is present, it will have low coverage, and the breadth of the MAG covered will be below the threshold of 50%. Indeed, as can be seen in Supplementary Fig. S5, there is a clear gap between samples where a given species is detected (in large circles that are outlined in black) or not (in small circles with no outline and usually red). We inspected such plots for all the species represented in the database and visually verified that the breadth of coverage cut-offs we used were valid across species. Plots of *Snodgrassella* are shown in Supplementary Fig. S5. All remaining plots are included with the dataset found in the Zenodo repository (<https://zenodo.org/doi/10.5281/zenodo.13732977>).

It now reads as: “[...] species was considered present if at least 50% of its genome (comprising all scaffolds) was covered by the reads. The justification for the choice of this threshold is exemplified for several *Snodgrassella* species in **Supplementary Fig. S5**: briefly: there is a clear gap between samples where a given species is detected (in large green circles) or not (in small red circles). We inspected such plots for all the species represented in the database and visually verified that the breath coverage we

used (50%) was valid across species. These plots are included with the dataset found in the Zenodo repository [...]"

Supplementary Fig. S5. Read mapping to species-representative MAGs. Read mapping (for *Snodgrassella* species across samples as an example) to closely related species. Different *Snodgrassella* species are represented in each panel. The x-axis in each panel represents samples grouped by host species (separated by vertical dotted lines), and the y-axis is the number of read-pairs mapping to the focal species in each sample (horizontal dotted lines represent different depths). The size of the points represents the breadth of the species-representative MAG covered in the sample and the color of the range into which its coverage falls. Points outlined with a black line correspond to samples in which the respective species was considered present according to thresholds used in the pipeline.

Additionally, I tried to run the pipeline, but the raw data is not accessible because the provided link (<https://zenodo.org/doi/10.5281/zenodo.13732977>) for data deposit is invalid. Thus, it is not possible to validate the code.

2.3 - Reply: It is unclear to us why the reviewer could not access our pipeline. The provided link, as copied by the reviewer above, is working for us. Also, the other reviewers were able to review our code with the provided links.

Reviewer #2 (Remarks on code availability):

The raw data is inaccessible because the provided link (<https://zenodo.org/doi/10.5281/zenodo.13732977>) for data deposit is invalid. Thus, it is not possible to validate the code.

2.4 - Reply: The raw data is deposited under Project ID PRJNA1157353 (<https://www.ncbi.nlm.nih.gov/bioproject/PRJNA1157353/>) on NCBI SRA as described in the Data Availability section.

The Zenodo repository (<https://zenodo.org/doi/10.5281/zenodo.13732977>) archives the code found in GitHub (<https://github.com/Aiswarya-prasad/honeybee-cross-species-metagenomics>) along with several intermediate files. Upon clicking on the link above, we find that it works as expected. The following page contains links to .zip files containing several intermediate files. We confirmed that the link works with other colleagues not involved in the study.

Published September 11, 2024 | Version v5

Computational notebook  Open

Files and Code Repository: Symbiont loss and gain, rather than co-diversification shapes honeybee gut microbiota diversity and function

Aiswarya¹

Show affiliations

The code repository accompanying the manuscript (*pre-print*), "Symbiont loss and gain, rather than co-diversification shapes honeybee gut microbiota diversity and function". When manuscript revisions are made, new releases and corresponding versions of the Zenodo (10.5281/zenodo.13732977) archive will be made.

Abstract

Studying gut microbiota evolution across animals is crucial for understanding symbiotic interactions but is hampered by the lack of high-resolution genomic data. Honeybees, with their specialized gut microbiota and well-known ecology, offer an ideal system to study this evolution. Using shotgun metagenomics on 200 honeybee workers from five species, we recovered thousands of metagenome-assembled genomes, identifying several novel bacterial species. While microbial communities were mostly host-specific, we found both specialist and generalist bacteria, even among closely related species, with notable variation between host species. Some generalists emerged host-specific only at the strain level, suggesting recent host switches. Unexpectedly, we found no evidence of codiversification between hosts and symbionts. Instead, symbiont gains, losses, and replacements led to functional differences, such as the ability to degrade pollen-derived pectin. Our results provide new insights into gut microbiota evolution and uncover the functional potential of the previously underexplored gut microbiota of these important pollinators.

Files

Name	Size	
README.md		Files (44.7 GB)		
05_assembly.zip md5:61f92dfc9c4538f011b04dabbe42d44b	14.5 GB	 Preview  Download
06_metagenomicORFs.zip md5:250cc26ce8afda3216cde66a3793db83a	2.8 GB	 Preview  Download
09_MAGs_collection.zip md5:713a889e94af8d74f285241b3c4761e4	12.1 GB	 Preview  Download
11_phylogenies.zip md5:480ba2af97d3db74b96900b752bc9	1.0 GB	 Preview  Download
Aiswarya-prasad/honeybee-cross-species-metagenomics-v1.0.3.zip md5:e73cc32b510aaa3ab10ae23c444d374e	44.8 MB	 Preview  Download
figures.zip md5:146d7dc8686b28a35ba3440a49811a67	14.2 GB	 Preview  Download
README.md md5:4fb21249de1ac33ccaac8c8a8f383ade44	279 Bytes	 Preview  Download

Reviewer #3 (Remarks to the Author):

The authors have conducted a very interesting study characterizing the microbiome of five *Apis* species: *A. cerana*, *A. dorsata*, *A. florea*, *A. andreniformis*, and the well-known *A. mellifera*, which serves as a key model system for microbiome research. While the microbiomes of *A. mellifera* and *A. cerana* have been previously characterized by the same research group also using shotgun sequencing (L91), studies on the remaining three species are sparse, likely due to their status as non-managed species. The authors analyzed a total of 195 samples (from 40 individual hives) and identified 150 dereplicated metagenome-assembled genomes (MAGs) of

medium to high quality. Community and functional analyses were conducted, and by leveraging the phylogenetic relatedness of these species within the monophyletic Apini clade, the study also addresses a critical question in microbiome research: the potential for co-diversification of microbiomes with their hosts. The authors state (L102) that "...ancient gut microbiota-host interactions can be maintained in the absence of co-diversification", which I comment below. The figures are thoughtfully designed to highlight the main findings, and the manuscript is clear and well-written, although a few additional details in the introduction would be helpful. The methods and software used were appropriately applied, with well-documented code made available.

The novelty of this study is the in-depth characterization of three non-model *Apis* species using shotgun metagenomics, providing greater resolution to investigate the evolution of the honey bee microbiome. Additionally, this is the first study to test for co-diversification in honeybees without relying on 16S rRNA data. These findings can certainly contribute to the broader microbiome field, which so far has observed co-diversification patterns in other systems, such as the microbiome of primates and humans. In this study, however, the authors suggest that the gains and losses of bacterial species appear to be among the most significant drivers of honeybees' microbiome composition, a pattern also observed and suggested in previous studies on corbiculate bees (*Apidae* clade) and stingless bees (*Meliponini* clade) (doi: 10.1126/sciadv.1600513, 10.1128/mbio.02317-21).

3.1 - Reply: We thank the reviewer for their detailed review of our study and constructive feedback. We are also thankful for their recognition of the novelty of our study and the validity of our approach.

Major comments

My only concern relates to the main conclusions of the manuscript stating that symbiont loss and gain—rather than co-diversification—shapes honeybee gut microbiota diversity and function. The authors used genus-level phylogenies to test for co-diversification (L200) because the divergence times within genera are roughly equivalent to those in the *Apis* genus (*Apini* tribe). However, this implies that diverged bacterial species and strains could exist within the *Apini* tribe. Thus, a robust test for co-diversification would necessitate a well-resolved symbiont phylogeny, particularly at the strain level. Ideally, collecting strain isolates from prevalent genera across different bee species and conducting transplant experiments would enhance the analysis. Nonetheless, given the challenges of working with non-model honeybee species, metagenomics remains a gold standard approach. With that in mind, would it not provide better resolution (although still not the ideal scenario) to dereplicate at 99% average nucleotide identity (ANI) and perform co-diversification tests at a finer scale? Discussion on dereplication in reference 10.1128/mSphere.00971-19.

3.2 - Reply: We thank the reviewer for emphasizing the importance of using genus-level phylogenies that are well-resolved within a genus. We agree that the signals of co-diversification are most thoroughly tested with well-resolved phylogenies, including several related MAGs per species that can represent genomes of different strains.

This is exactly what we have done, but we realize this was not clearly explained in the previous version of our manuscript. We did indeed use all the medium and high-quality MAGs (not just the representative MAGs of each dereplicated species cluster) in our phylogenies. This means that each clade of the phylogeny contains several MAGs that are >95% similar in ANI and some that are nearly identical (equivalent to dereplicating at 100% ANI), providing very high, i.e., strain-level resolution. We have now updated the Methods section (Lines 615 – 616) in order to ensure that this information is clearly conveyed.

As mentioned in the manuscript, previous studies in other model systems suggest that only a small percentage of clades (taxa) within a microbial community would co-diversify with their host. Nevertheless, this is an important finding as it highlights an evolutionary mechanism driving microbiome evolution, and thus deserves further exploration. Looking at Table S6 (L215), it appears that significant co-diversifying nodes were identified at the various thresholds used. If I have misinterpreted this, could you please clarify the table? Including the expected co-diversification pattern by chance might be helpful, as I was unable to find this comparison in the tables or figures. If only a single bacterial clade appears to be co-diversifying, it may be valuable to use the most complete MAGs to test for gene functions that are significantly overrepresented in this co-diversifying clade compared to non-diversifying clades, and to calculate gene dN/dS ratios, for example.

3.3 - Reply: We thank the reviewer for bringing to our attention the lack of clarity in conveying the number of co-diversifying clades. We agree that our presentations of this pivotal aspect of the analysis lacked clarity.

We have modified the manuscript in several ways to improve clarity:

1/ We have updated the table legend of Supplementary Table S6, which presents the number of significant co-diversifying nodes along with null expectations. Briefly, in Supplementary Table S6, the first column under each section (relaxed, strict and medium) is named “# significant”. This represents the number of nodes passing the respective threshold using the Hommola test on the true host and symbiont phylogenies. The next two columns list the median and standard deviation of the number of nodes passing the threshold in second-order permutation tests randomizing the node labels N=100 times. In other words, these are the median and standard deviation of the number of significant clades expected only by chance. One can see that the observed number of significant nodes is often not higher than what is expected by chance, suggesting that we cannot reject the hypothesis that there is no co-phylogenetic signal.

2/ To better illustrate the method we used, we have added a schematic including a simple example to the Supplementary material that more intuitively explains the co-diversification test we used (posted below and added to the revised manuscript as Supplementary Fig. S20).

Clade X

Actual comparison

(node-by-node Hommola test)

Supplementary Fig. S20. Schematic of co-diversification testing approach. This schematic describes the approach used to test for co-diversification. For a given clade (in the case of our trees, genus), the nodes that pass filtering thresholds (eg., here, at least 4 MAGs and at least 3 host species represented) are identified. The Hommola cospeciation test (as implemented in the Python package *skbio*) is applied to each identified node in the tree, considering, in each case, the host matrix (H) and symbiont matrix (S) containing tip-tip distances from phylogeny and the host-symbiont interactions list (I). Subsequently, the number of significant nodes in the tree can be found. Next, to establish a false-discovery (FDR) rate, an expectation for the number of significant nodes can be computed using second-order permutation tests with tip labels of the host tree shuffled in different permutations. Using this, an FDR can be established to indicate whether the significant node(s) identified in the actual comparison is(are) likely to be false positive(s). The black horizontal line on the final plot on the right represents the total number of nodes tested, the grey circles are the number of significant nodes across second-order permutation tests with their size indicating the frequency, and the horizontal red line is the number of significant nodes in the actual comparison.

3/ We have now included an additional panel in the main figure (Fig. 3B), illustrating the number of significant nodes detected along with the number expected only by chance. The number of significant nodes was never higher than expected by chance for any of the genera tested with strict or medium thresholds (Fig 3B, Supplementary Table S6).

4/ We have added tanglegrams (as requested by reviewer #4, see comment #4.1) for the nodes with the strongest co-phylogenetic signal (Supplementary Fig. S12). Visual inspection of the trees descending from these nodes crossing the “relaxed” threshold further lends confidence to our conclusion that they are likely false positives. For example, several nodes show congruence but only cover three different host species, with one of them being represented by one or two MAGs but others with several, which can lead to increased R and significant p-values. Several other nodes do not show congruence with the host phylogeny. This is in stark contrast with the examples of co-diversifying nodes

found in Primates (Moeller *et al.*, 2016; Sanders *et al.*, 2023; Rühlemann *et al.*, 2024) (see also a response to reviewer #4, 4.1). Tanglegrams of nodes marked significant under different thresholds are plotted in Supplementary Fig. S13 (reproduced below) and show no strong congruence with the host phylogeny except when a strict R-value threshold (>0.75) is used. We have also added the following lines to the manuscript to discuss how our results can be viewed in comparison to other systems where co-diversification was seen. Lines 386 – 392 now read: “Our results markedly contrast with co-diversification findings in mammals and, in particular, hominids⁶⁻¹¹. In those studies, using a similar approach, the ratio of the number of significantly co-diversifying clades observed and expected only by chance was one order of magnitude higher than here (~ 12 vs. ~ 1.2). In addition, clear examples of highly congruent phylogenies can be found in these studies in hominids, which is not the case here.”

Tree from example node significant under:

Strict threshold ($r = 0.77, p = 0.002$)

Medium threshold ($r = 0.79, p = 0.021$)

Relaxed threshold ($r = 0.29, p = 0.006$)

Supplementary Fig. S13. Tanglegrams of symbiont and host phylogeny of selected nodes.

Subtrees of selected nodes of varying levels of significance. Each tip of the tree is labeled with the corresponding species and MAG name. The colors indicate the host species from which it was recovered.

We have now improved the explanation of our approach and further justified it here. Calculating dn/ds ratio, while very exciting, is probably beyond the scope of this already dense manuscript.

Minor comments

- L83: I mostly recall studies suggesting gain and loss of microbial associations and discussing the potential for co-diversification. Add references for “Co-diversification has been suggested as the primary mode of evolution” to support this statement.

3.4 - Reply: We agree that more references are needed to substantiate this claim. We have now added references that were based on 16S rRNA amplicon sequence-based trees (Koch *et al.*, 2013; Kwong *et al.*, 2017). More generally, co-diversification seems to be assumed in the field even without referring to a formal test, e.g., in (Engel, Martinson and Moran, 2012; Wu *et al.*, 2021; Luo, Zhang and Zhou, 2024). We believe this reflects the lack of clarity in the field so far and hope that our results provide a clearer view.

In addition, we have also replaced “the primary mode of” with “an important mode of” in the text (Line 88).

- L96: It would be helpful for the reader if the authors could better contextualize the host species, including information on where they occur and the main differences in their biology. This additional context would also enhance the discussion later in the manuscript, such as in L301.

3.5 - Reply: We have added information about host species distribution and ecology to the introduction section (Lines 107 – 115).

It now reads as: “worker bees of the five most prevalent species of honeybees described so far, and spanning all three major clades of honeybees, namely Micrapis (dwarf honeybee): *A. florea* and *A. andreniformis*, Megapis (giant honeybee): *A. dorsata*, Apis: *A. mellifera* and *A. cerana*. While *A. mellifera* is globally distributed due to human introduction, the other honeybee species are mainly found across Asia²¹. All five honeybee species share several traits, such as a complex social lifestyle and a diet mainly consisting of processed pollen and nectar. However, they differ in certain key aspects, including body and colony size, nesting habit (open versus cavity nesting) and migrating ability⁴².”

- L110: Were all samples collected from wild nests? Make sure this detail is also included in the methods section.

3.6 - Reply: Most of the colonies were wild except the *Apis mellifera* colonies. This information is now added to the methods section (Lines 441 – 444).

- L113 and Fig S2: Double-check number “1959” or “1958” MAGs.

3.7 - Reply: We thank the reviewer for catching this oversight. There are indeed a total of 1,959 MAGs of good (medium or high) quality. The discrepancy arose because the plot in Supplementary Fig. S2 includes only MAGs with a size less than 5M to exclude spurious MAGs, and this results in the exclusion of MAG A5-4_6 which has a size of 5.44M, even though it is not likely a spurious

binned MAG, just a species with a larger than average genome. We have updated the figure to also include this MAG, and the updated code and data repositories will reflect these changes.

- Fig S8: It appears that some bacterial species are represented with the same color, making them difficult to distinguish.

3.8 - Reply: Since it is challenging to find a discrete color palette to distinguish 10+ colors for genera with too many species, we have added thicker separating lines to aid in distinguishing bars representing different species (Supplementary Fig. S10).

- L164: You introduce the term 'phylosymbiosis' here, which was not mentioned in the introduction. It can be helpful to clearly define and differentiate this term from co-diversification and co-evolution in the introduction. Distinctions are discussed in studies like doi: 10.1111/jeb.14221.

3.9 - Reply: We have mentioned phylosymbiosis in the introduction (Line 44) and noted that it is distinct from co-diversification with the above reference cited (Line 51 – 52).

It now reads as: “In some animal groups, it has also been observed that more closely related host species harbor more similar gut microbiota in composition, in a pattern referred to as phylosymbiosis [...] only a small fraction of the analyzed bacteria showed signs of co-diversification (a pattern different from phylosymbiosis) [...]”

- Fig 2B and C: Reading the methods, it was not entirely clear whether the input data for host-specific analysis was at the strain level and how it was used here. Does inStrain provide information on whether specific MAGs with particular SNPs are present in a sample?

3.10 - Reply: The host specificity analysis (Fig. 2A) was at the species level, but we further tested within each bacterial species for host-specificity at the strain level (Fig. 2B and Fig. 2C). InStrain provides information about which SNPs are present on the scaffolds of each species-representative MAG of the database within each sample. The strain-level profile is obtained per bacterial species using inStrain, and NMDS plots like those shown in Fig. 2C were generated using popANI values calculated by inStrain for each bacterial species between each pair of samples. Fig. 2B was made by summarizing each species, whether it was host-specific i.e., the PERMANOVA test on the popANI data suggested host-specific distribution ($p < 0.01$) or not. We have included more details on this in the methods section (Lines 549 – 551, 565 – 568) as well as in the figure legend.

It now reads as: “Host specificity was calculated for each bacterial species as the Rohde’s index of specificity³⁹ using the prevalence of the species across hosts [...] Finally, each of the bacterial species was either considered host-specific, if they were detected in only one host species, or shared if not. Shared species were considered host-specific at the strain level if samples showed significant clustering by host species ($p < 0.01$) according to a PERMANOVA test.”

- L230: Figure “Fig. 4A”.

3.11 - Reply: This has been corrected.

- L235: Make sure to cite Fig. 4B here. In fact, the entire paragraph could include citations to Fig. 4 wherever relevant data is discussed.

3.12 - Reply: We have added appropriate cross-references.

- L303: Clarify what you mean by “specific ecological niches” and add a reference. Provide more details on whether this relates to diet, nesting locations, or other factors.

3.13 - Reply: Specialized ecological niches can be defined as a narrow range of conditions and resources that can be exploited by an organism. We have also added examples and references (Lines 365 – 379).

It now reads as: “specific ecological niches⁵⁵ (a narrow range of conditions or resources that an organism can exploit) than the other bee species, facilitating the colonization and maintenance of additional ‘specialist’ bacteria. This is consistent with the observation that *A. cerana* and the two dwarf bees harbor significantly fewer bacterial species per individual bee than *A. dorsata* and *A. mellifera*. Several different factors could contribute to these patterns. Ecological niches in the host gut can vary even among closely related host species due to differences in host physiology, diet, or gut structure. For example, the host-specific colonization of *S. alvi* of *A. mellifera* may be supported by specific organic acids secreted by the host into the gut lumen and utilized by this bacterium⁵⁵. Slight differences in dietary preferences, e.g the amount or type of pollen, may allow different combinations of species to co-exist in one host species compared to another⁵⁷. Variation in body and colony size between *Apis* species may determine the carrying capacity and hence bacterial diversity in the gut, as proposed previously^{26,40}. Another possibility is that host species may exert different extents of control over their gut microbiota, for examples, e.g. via the immune system, as suggested in other animals⁵⁸.

- L342: Italicize *A. dorsata*.

3.14 - Reply: This has been corrected.

Reviewer #4 (Remarks to the Author):

Prasad et al 2024 – Reviewer comments

The authors performed shotgun sequencing on 5 honey bee species hind guts. They generated a dataset to show which bacterial species were present in which honey bee species and compared diversity across hosts. The authors noted the presence of specialist vs generalist bacterial species. They also found in some cases, for shared species, host-specificity exists at the strain level. The authors constructed core genome phylogenies. They found novel clades of bacteria, many in *Apis dorsata*.

The authors also tested for co-diversification (Hommola cospeciation test) in 9 genera. They conclude co-diversification did not occur in the majority of species/host pairs. In phylogenies, host specific clades are interspersed by generalist clades. The authors also looked at functional differences in the microbiota across the bee species. They looked at KO functional groups of the microbiota and found them to be distinct across all host species. Some host specific genera (eg, *Dysgonomonas*) had their own clusters, suggesting unique function within a host. A few genera (*Saezia* and *Entomomonas*) clustered with unrelated genera (eg, *Snodgrassella*), suggesting functional replacement. In the case of *Dysgonomonas*, the authors provide evidence

that this genera helps to degrade plant fiber

Overall, the authors have performed a comprehensive study on bee microbiome diversity in different honey bee species. The experimentation and initial analysis appear to be sound. However, the interpretation of the analysis and cutoffs used seem to be leading to the opposite conclusion that I draw when I look at these data (ie, that co-diversification is occurring, with potentially other evolutionary drivers contributing). See comments below.

This work has the potential to be impactful for evolution and host specificity of bee microbiome members, provided the conclusions are modified. Specifically, the data show co-diversification occurs, but is not the sole driver of evolution in bee microbiome members (as evidenced by weak-moderate correlations for some nodes).

Major comments

1. The claim that “honeybees and their gut bacteria did not codiversify” (line 217-218) is too strong. The data (tables S5 and S6) show a large number of the bacteria test have significant p values. The authors use a cutoff of $r > .75$ (as in Sanders et al 2024), but this omits significant correlations that are still weak/moderate/and at the lower end of strong (<https://sphweb.bumc.bu.edu/otlt/MPH-Modules/PH717-QuantCore/PH717-Module9-Correlation-Regression/PH717-Module9-Correlation-Regression4.html>). For example, 52/103 nodes with significance (table s5) have an r of .4 or greater, indicating a moderate-strong correlation. Using this lower r, one would conclude that co-diversification is present in over half of the relationships tested, at a moderate amount. If an R value of $> .2$ (weak correlation) is used, then 88/103 nodes display co-diversification, with weak correlation.

I believe a more appropriate conclusion would be that since most nodes have weak or moderate correlations, the data suggest co-diversification is occurring, but is not the only process driving evolution. Ie, another process like host-switching (as alluded to) could play a larger role in driving evolution in nodes with low R values.

I would recommend to flesh this section of the results out by including some tanglegrams and describe the bins of nodes in strong, moderate, and weak correlation, as done above.

4.1 - Reply: We thank the reviewer for their detailed thoughts on this section. We agree that our wording was too strong, although we still believe that co-phylogenetic patterns are extremely (and maybe surprisingly) weak in *Apis*, at least compared to other gut microbiomes. We think the point made by the reviewer stems from three main reasons:

1/ Strong wording in our previous manuscript, e.g., the use of “no” co-diversification. We have softened the text in several lines, as explained below.

2/ A lack of clarity in the methods of the previous manuscript regarding the set of tests that were carried out. This was also a comment from reviewer #3. We have included additional text and new figures to better explain our approach. (see below).

3/ A lack of explicit visual and quantitative outputs illustrating the very low amount of co-phylogenetic pattern, even for clades with large r values. We have now produced multiple new figures to better illustrate this point (see below).

Overall, this comment helped us expand our manuscript and better explain and present our results. We feel this part of our manuscript is now much improved and stronger. Please find below a more detailed response for each of the three points above.

1/ Wording.

We rephrase some of the strong statements from the previous manuscript in several lines. For example, we replace “honeybees and their gut bacteria did not codiversify” by the following in Lines 256 – 258: “[...] we show that the strength of the signal of co-diversification between honeybees and their gut bacteria is not higher than expected by chance despite high host specificity”

2/ Clarification of our approach.

We have now produced a new figure to explain the overall approach, as also requested by reviewer 3 (posted below and included as supplementary Fig. S20) See also related response 3.3 to reviewer 3 for more details on this point.

3/ Better representation of the results and interpretation of statistical results.

Here, we first clarify the reasoning behind our cut-offs and the decision not to consider R -values below 0.75, even when p -values are significant. Post hoc analysis conducted by (Sanders *et al.*, 2023) provides useful insights for determining appropriate R -value and p -value thresholds in co-diversification tests. Low R -values accompanied by significant (low) p -values can arise due to a large number of MAGs (metagenome-assembled genomes) within a subtree or when there are fewer host species represented by the MAGs (Sanders *et al.*, 2023). Therefore, careful interpretation of the Mantel R -value is required. For example, a node with a p -value > 0.01 and an R -value indicative of weak correlation does not necessarily reflect “weak co-diversification”. Instead, such results often stem from having a large number of MAGs of each bacterial species (which cluster by host) and/or a few host species (three rather than four or five) represented within the subtree at that node. This becomes clearer when comparing subtrees detected under different thresholds:

- Strict and Medium thresholds impose a p -value cut-off and require $R > 0.75$.
- Relaxed thresholds apply only the p -value cut-off.

To visualize nodes that pass various thresholds, we have now included tanglegrams of nodes identified as significant under various thresholds as suggested by the reviewer, showing significant nodes identified from each of the genera (new Supplementary Fig. S12) and another to show an example of nodes identified under various thresholds (Supplementary Fig. S13, also posted in the reply to reviewer #3 above).

As seen in Supplementary Fig. S13, small R values, despite p -values < 0.01 , do not represent nodes that are largely congruent with host phylogeny. In

particular, the signal seems to stem only from the fact that bacteria associated with the two dwarf bees are closely related. Hence, we consider nodes significant only when both the R-value and p-value cross thresholds. Given that relying on these thresholds could still lead to false positives, a FDR (false discovery rate) can be established. To do this, we use the approach laid out in (Moeller *et al.*, 2023), described in the schematic shown above (and included as Supplementary Fig. S20 in the revised manuscript) and referenced in the methods section Line 616.

We established the number of significant nodes to be expected by chance, carrying out the node-by-node comparison (Hommola cospeciation test, see schematic in Supplementary Fig. S20) between the symbiont tree and host tree, and repeated that comparison by replacing the host tree with another tree with host node labels shuffled (second-order permutation, N=100). Within the actual comparison, there are few nodes that cross the threshold; however, based on the results from the second-order permutation tests using shuffled labels, they are most likely false positives. For example, consider the *Lactobacillus* clade. Under relaxed thresholds, there are 24 significant nodes in the actual comparison and 22 (median) across the permutations. The False discovery rate would be (Expected False Positives/Total discoveries) $22/24 * 100 = 91.67\%$. This means that all these nodes but two are most likely false positives. Under strict thresholds, the FDR is 100%, meaning that the nodes detected are most likely false positives. Hence, we conclude that there is no more co-diversification than expected by chance. Across genera, regardless of the R- and p-value thresholds used, we do not find more co-diversification in the actual comparison compared to the ones with node labels shuffled. To clearly convey this to the reader, we have added a new panel to Fig. 3.

That being said, we feel it is still important to acknowledge that some clades show a very weak signal of co-phylogeny, which led us to soften our interpretation, e.g., Lines 253 – 256: “Closer inspection of the subtrees from these nodes showing the highest signals of potential co-diversification for each genus showed little congruence with host phylogeny (Supplementary Fig. S12, Supplementary Fig. S13).”

Finally, it is important to bear in mind that co-phylogeny does not imply co-diversification. Co-diversification is only supported if the time scale of the host and symbiont nodes match as well. From the tanglegrams (Supplementary Fig. S12), there is little support for either pattern.

In addition to documenting the magnitude of co-phylogenetic patterns in our system, we have now also compared it with those found in other systems to understand whether co-phylogenetic patterns are more prevalent, less prevalent or similar across animals. Using the approach described in the schematic above, co-diversification has been detected in primates in (Sanders *et al.*, 2023). In that case, using the same strict threshold as us, they observed 206/1,616 significant nodes as compared to the number expected by chance, 16/1,616 established from second-order permutation tests. This is orders of magnitude higher than what we have found in our study (16/192, while 13/192 was expected by chance). They also demonstrated strict congruence with host phylogenies in subtrees at identified significant nodes. Surprisingly, this is not the case with our datasets (Fig. 3B and Supplementary Fig. S12). We added

text to emphasize this important comparison of co-phylogenetic strength with primates. Lines 386 – 392 now read: “Our results markedly contrast with co-diversification findings in mammals and, in particular, hominids⁶⁻¹¹. In those studies, using a similar approach, the ratio of the number of significantly co-diversifying clades observed and expected only by chance was one order of magnitude higher than here (~12 vs. ~1.2). In addition, clear examples of highly congruent phylogenies can be found in these studies in hominids, which is not the case here.”

In summary, while our initial expectation was that there would be co-phylogenetic patterns across honeybees, we can only conclude - based on our results and the comparison with previous studies - that there is very little evidence that co-diversification has driven the evolution of the major core members of the bee microbiota. Overall, we hope that the additional explanation and figures will more clearly convey our results and more effectively support our conclusions.

2. Related to comment #1, the language throughout, including the title, should be softened to convey that other forces, in addition to co-diversification, shape the evolution of bee microbes

4.2 - Reply: While we hope that the additional figures and explanation support our conclusion that there is no more co-diversification than expected by chance, we have softened the interpretation, for example

In the abstract,

Line 30: “Unexpectedly, we found little evidence of co-diversification between hosts and symbionts.”

In the results,

Lines 253 – 256: “Closer inspection of the subtrees from these nodes showing the highest signals of potential co-diversification for each genus showed little congruence with host phylogeny (Supplementary Fig. S12, Supplementary Fig. S13).”

Lines 247 – 249: “In summary, we show that the strength of the signal of co-diversification between honeybees and their gut bacteria is not higher than expected by chance despite high host specificity”

Concerning the title, we feel that given the co-phylogenetic signal is **not higher than expected by chance**, our title is not too strong and faithfully reflects our results. We have thus decided to keep it as such.

3. The language on host switching as a driver for evolution is also too strong (Eg, line 195-196, line 324). Host switching can be proposed as a possible additional factor driving evolution.

4.3 - Reply: - We acknowledge that host switches are one of several factors driving evolution and have hence mentioned symbiont gains and losses alongside host switches. As we do not observe more co-diversification than expected by chance, host switches, gain and losses likely represent the most predominant processes.

e.g., Lines 397 – 400: “Overall, our results show that ancient and specific gut microbiota-host interactions can be maintained in the absence of prevalent co-diversification^{60,61}. Instead, host switches, gains and losses between related species could have been frequent, suggesting that they govern compositional changes in gut microbiota over evolutionary time.”

As explained in the response above, we do not observe more co-diversification than expected by chance, yet there are clear examples of host-specific clades. This pattern has actually been coined as the “parasite paradox” in the co-evolution literature: see e.g., (Agosta, Janz and Brooks, 2010; Janz, 2011). We now refer to this in the discussion (Line 397).

In addition, there are multiple potential cases of host switches that warrant further investigation. We feel the discussion on this point was lacking in the previous version of our manuscript, so we have expanded on this.

The host-specific clades that appear to be maintained through processes distinct from co-diversification represent excellent candidates for future studies aimed at elucidating the ecological mechanisms underlying host specificity. Consequently, we believe that exploring the processes governing the loss, gain, and switching of microbiota members across host species is a valuable direction for future research and offers intriguing insights into the broader dynamics of host-microbiota interactions.

This is now mentioned in Lines 411 – 413: “To better understand the evolutionary dynamics of the bee gut microbiota, we suggest that it will be key for future research to identify the rate and determinants of host switches, gain and losses.”

4. Figure 4B shows distinct clusters of bacterial KO annotation profiles based on host species. I.e, bacteria cluster together based on host despite being in different clades. This seems to also provide evidence for co-diversification between host and microbiota. The authors might consider reframing how they discuss these data in the results to indicate this point.

It might be helpful to see Figure 4B with another dimension – bacterial genus (perhaps use shape host species and color for bacterial genus). It is unclear if bacteria cluster better within host species or bacterial genus. If the former, than this can be taken as evidence in favor of co-diversification.

4.4 - Reply: We thank the reviewer for pointing out the lack of clarity in Fig. 4B. In this figure, each point represents a sample and, as a result, the collection of all KOs across all MAGs in the sample. In Fig. 4C, on the other hand, each point represents a MAG and hence shows that MAGs KO profiles cluster by bacterial genus, not host species. We have clarified this by modifying the figure legend.

Fig. 4C shows that (i) MAGs from the same bacterial genus cluster together in terms of their functional profile; (ii) some unrelated genera have similar functional profiles (and hence may occupy similar niches in the gut); (iii) some functional profiles are conserved across different honeybee species (e.g. those of *Lactobacillus* and *Bifidobacterium*), while others are host-specific or occupied by different genera in different hosts. This figure hence expands on the pattern seen in Fig. 4B as suggested by the reviewer.

We have added a clearer explanation to the legend to improve the clarity of the relevant figure panels: “(B) PCoA of KO family composition across all samples was visualized using the Robust Aitchison distance on a matrix of reads per kilobase per million (RPKM) summed across genes belonging to the respective KO family. Each point on the plot represents an individual honeybee sample and the color the host species identity of the sample. PERMANOVA test indicated on the plot tests

the effect of host species identity. (C) PCoA of all KO families from genes annotated within each MAGs across all samples using Robust-Aitchison distance on the RPKM matrix of KOs. Each point represents a bacterial MAG colored by the bacterial genus it belongs to. MAGs obtained from different host species were separated across panels for clearer visualization. The axes across the 5 panels have been maintained to the same limits. So, the panels can be superposed. from each host species plotted on the same scale across host species using the Robust-Aitchison distance on the RPKM matrix of KOs of MAGs such that each point on the plot represents a MAG recovered from an individual bee.”

Specific comments

1. Line 129-130: Be specific in the text that this study increases MAGs for species that have been undersurveyed to this point (specifically, *A dorsata* and *A florea*).

4.5 – Reply: This has been updated in Lines 151 – 153: “The established collection of MAGs massively expands the catalog of bacterial genomes from the gut environment of these underexplored honeybee species.”

2. Line 226: define KO identifier

4.6 – Reply: This has been updated.

3. Line 230: Fig 2A should be Fig 4A

4.7 – Reply: This has been corrected.

4. Lines 232-Lines 247: Figs 4B and 4C should be cited as they are discussed in the text/

4.8 – Reply: Appropriate cross-references added.

5. Line 242-243: Can you provide any additional information about how the KO annotations in *Saezia* and *Entomomonas* are similar to *Snodgrassella*? Ie, any hypotheses on what function might they be replacing?

4.9 – Reply: We thank the reviewer for raising this point. We have carried out additional analysis in the new version of the manuscript, as outlined below.

Specifically, we have used MinPath (see updated methods Lines 642 – 644) to infer the presence of pathways in each MAG using the KOs encoded within. Further, we used MaAsLin 2 (see updated methods Lines 644 – 648), which was developed to assess multivariable associations of microbial community features with complex metadata. With the presence-absence matrix of pathways as the dataset and ‘genus’ as the explanatory variable, we saw that the three genera mentioned above showed significant association with pathways under carbohydrate metabolism. The heatmap (Supplementary Fig. S15) shows how the pattern might arise from similarities and differences in carbohydrate metabolism. Specifically, these three genera contain a TCA cycle while lacking pathways for sugar utilization, including Pentose and glucuronate conversions, Fructose and mannose metabolism, Galactose metabolism and Ascorbate and aldarate metabolism, and starch and sucrose metabolism, which are all present in most other MAGs.

We have now (i) modified the method to incorporate this new analysis Lines 642 - 648, (ii) added Supplementary Fig. S15 to demonstrate this further, (iii)

added new data and code to the Zenodo and Github repositories, (iv) and added, “Closer inspection of their metabolic profiles indicated that both *Saezia* and *Entomomonas* like *Snodgrassella* lack major carbohydrate metabolism pathways (Supplementary Fig. S15), suggesting potential functional replacements.” in Lines 286 - 289.

6. Line 258-265: There is a logic jump here. A dorsata contains multiple bacterial genera that contain pectin/cellulose degrading enzymes. *Dysgonomonas* also has polysaccharide degrading enzymes. It is unclear, as written, that the PULs in *Dysgonomonas* are expected to degrade unique polysaccharides. If there is literature suggesting PULs have a range of substrate specificity and presence of a unique PUL suggests a unique polysaccharide substrate, then this should be cited. The phrase “expands the functional capability to degrade plant-derived fibers” needs more explanation.

4.10 – Reply: We thank the reviewer for highlighting the lack of clarity in reasoning in this section. We have added Supplementary Fig. S18 (referenced in Line 299) to clearly illustrate the distribution of CAZyme families across bacterial species. From here, it is clear that *Dysgonomonas* encodes several CAZyme families, which are not found in other genera. This suggests unique enzymatic capacities, e.g., to cleave specific covalent bounds in the heteropolysaccharide pectin. However, since this is based on annotation, we cannot – at this point – exclude that some of these enzyme functions may be redundant with pectinases found in other genera. However, these enzymatic capabilities have so far mostly been attributed to *Gilliamella* in the bee microbiota (Engel, Martinson and Moran, 2012; Zheng *et al.*, 2019). So, it is highly interesting to see that in *A. dorsata*, another genus seems to contribute to the degradation of this highly abundant polymer in the pollen diet.

We have modified the sentence to “[...] suggesting that this community member contributes to the capability of this particular honeybee species to degrade plant-derived fibers (Fig. 4F)”.

Reviewer #4 (Remarks on code availability):

n/a

References

Agosta, S.J., Janz, N. and Brooks, D.R. (2010) ‘How specialists can be generalists: resolving the “parasite paradox” and implications for emerging infectious disease’, *Zoologia (Curitiba)*, 27, pp. 151–162. Available at: <https://doi.org/10.1590/S1984-46702010000200001>.

Brochet, S. *et al.* (2021) ‘Niche partitioning facilitates coexistence of closely related honey bee gut bacteria’, *eLife*, 10, p. e68583. Available at: <https://doi.org/10.7554/eLife.68583>.

Ellegaard, K.M. *et al.* (2020) ‘Vast Differences in Strain-Level Diversity in the Gut Microbiota of Two Closely Related Honey Bee Species’, *Current Biology*, 30(13), pp. 2520-2531.e7. Available at: <https://doi.org/10.1016/j.cub.2020.04.070>.

Engel, P., Martinson, V.G. and Moran, N.A. (2012) 'Functional diversity within the simple gut microbiota of the honey bee', *Proceedings of the National Academy of Sciences*, 109(27), pp. 11002–11007. Available at: <https://doi.org/10.1073/pnas.1202970109>.

Jain, C. *et al.* (2018) 'High throughput ANI analysis of 90K prokaryotic genomes reveals clear species boundaries', *Nature Communications*, 9(1), p. 5114. Available at: <https://doi.org/10.1038/s41467-018-07641-9>.

Janz, N. (2011) 'Ehrlich and Raven Revisited: Mechanisms Underlying Codiversification of Plants and Enemies', *Annual Review of Ecology, Evolution, and Systematics*, 42(Volume 42, 2011), pp. 71–89. Available at: <https://doi.org/10.1146/annurev-ecolsys-102710-145024>.

Koch, H. *et al.* (2013) 'Diversity and evolutionary patterns of bacterial gut associates of corbiculate bees', *Molecular Ecology*, 22(7), pp. 2028–2044. Available at: <https://doi.org/10.1111/mec.12209>.

Kwong, W.K. *et al.* (2017) 'Dynamic microbiome evolution in social bees', *Science Advances*, 3(3), p. e1600513. Available at: <https://doi.org/10.1126/sciadv.1600513>.

Luo, S., Zhang, X. and Zhou, X. (2024) 'Temporospatial dynamics and host specificity of honeybee gut bacteria', *Cell Reports*, 43(7), p. 114408. Available at: <https://doi.org/10.1016/j.celrep.2024.114408>.

Moeller, A.H. *et al.* (2016) 'Cospeciation of gut microbiota with hominids', *Science*, 353(6297), pp. 380–382. Available at: <https://doi.org/10.1126/science.aaf3951>.

Moeller, A.H. *et al.* (2023) 'Assessing co-diversification in host-associated microbiomes', *Journal of Evolutionary Biology*, n/a(n/a). Available at: <https://doi.org/10.1111/jeb.14221>.

Olm, M.R. *et al.* (2021) 'inStrain profiles population microdiversity from metagenomic data and sensitively detects shared microbial strains', *Nature Biotechnology*, 39(6), pp. 727–736. Available at: <https://doi.org/10.1038/s41587-020-00797-0>.

Quinn, A. *et al.* (2024) 'Host-derived organic acids enable gut colonization of the honey bee symbiont *Snodgrassella alvi*', *Nature Microbiology*, 9(2), pp. 477–489. Available at: <https://doi.org/10.1038/s41564-023-01572-y>.

Rühlemann, M.C. *et al.* (2024) 'Functional host-specific adaptation of the intestinal microbiome in hominids', *Nature Communications*, 15(1), p. 326. Available at: <https://doi.org/10.1038/s41467-023-44636-7>.

Sanders, J.G. *et al.* (2023) 'Widespread extinctions of co-diversified primate gut bacterial symbionts from humans', *Nature Microbiology*, 8(6), pp. 1039–1050. Available at: <https://doi.org/10.1038/s41564-023-01388-w>.

Wilde, J., Slack, E. and Foster, K.R. (2024) 'Host control of the microbiome: Mechanisms, evolution, and disease', *Science*, 385(6706), p. eadi3338. Available at: <https://doi.org/10.1126/science.adi3338>.

Wu, J. *et al.* (2021) 'Honey bee genetics shape the strain-level structure of gut microbiota in social transmission', *Microbiome*, 9(1), p. 225. Available at: <https://doi.org/10.1186/s40168-021-01174-y>.

Zheng, H. *et al.* (2019) 'Division of labor in honey bee gut microbiota for plant polysaccharide digestion', *Proceedings of the National Academy of Sciences*, 116(51), pp. 25909–25916. Available at: <https://doi.org/10.1073/pnas.1916224116>.

Our comments are in blue below. All cited references in the responses are listed at the end of the document. Line numbers correspond to marked-up manuscript (MS).

REVIEWER COMMENTS

Reviewer #2 (Remarks to the Author):

The authors partially addressed my concerns. In principle, this study collected bees of five species and sequenced them individually with Illumina paired reads. Then, the authors analyzed the microbial community with these reads. Several studies have confirmed bee species-specific microbiota and distinct microbes either at the species level or the strain level, so it is not surprising to find community variance among the five bee species. However, including the giant and dwarf honeybees improves our understanding of bee symbiosis.

If the authors conclude that a bee species gains a symbiont, it means this bee species didn't have this symbiont before, but has it now. It is the same for symbiont loss. There is no time series analyses for each species in this study. Thus, it is a single time point data. When referring to the host switch, who is the native host? Overall, robust evidence is needed to support this very important conclusion to make sure it is correct. The data didn't include any ancient symbionts, how can you determine that ancient microbiota-host interaction can be maintained in the absence of strict co-diversification?

Our response: Thank you for raising these points. Regarding the lack of a time-series sampling in our study, we would like to refer to our reviewer responses to the first round of revisions. We look at evolutionary time scales and not at ecological timescales when referring to symbiont gain and loss. While we agree that deep time series provided by fossils are important pieces of evidence to reconstruct evolutionary dynamics, this is not feasible with gut microbiota. So, to infer ancient symbiont loss and gains (which happened sporadically over the last few million years), we must rely on a comparative genomics approach. The argument made by reviewer #2 about the impossibility of inferring gains and losses without "time series" ignores more than a century of research in comparative biology. Inferring past evolutionary dynamics of symbionts and hosts (e.g., host gain and losses) using present-day data has been described in detail elsewhere (Johnson et al., 2003; de Vienne et al., 2013; Moran et al., 2019; Dismukes et al., 2022; Moeller et al., 2023). Hence, we did not include more detailed explanations in our text, which is already rather dense.

However, we have changed the statement about the 'maintenance of ancient microbiota-host interaction' at the end of the introduction and the discussion and replaced it.

L 122-124: "Overall, our results show that the evolution of specialized gut microbiota-host interactions in honeybees has been shaped by symbiont loss and gain, more so than by co-diversification"

The data heavily depends on the available genome assemblies when assigning the contigs to the microbial species. Symbionts in *A. mellifera* and *A. cerana* are well represented. While, giant and dwarf bees are underrepresented. The authors performed several analyses to reduce the bias, which is good. However, bias will still exist. The result interpretation must be careful, that is why I asked for species-specific PCR validation. I shared this with other reviewers and am concerned about the conclusion that there is no co-diversification.

Our response: The data does not depend on the isolated genomes when assigning contigs to microbial species bins. The assembly of the MAGs happens *de novo*, i.e., without the inclusion of information from the isolate genomes. So, it is not clear to us why the reviewer is concerned that the method should only work for those MAGs where we have isolated genomes. Since the assembly is done without the inclusion of the isolate genomes and contigs assigned to bins/MAGs *de novo*, there is no bias in this analysis due to under-represented species in terms of isolate genomes. Carrying out PCRs on selected genes may help verify the presence of these genes in a given sample, but it will not allow us to validate if the assemblies of the MAGs are correct. A PCR-based method is limited in taxonomic resolution and hence inferior to our approach (i.e., less comprehensive) for verifying the validity of our MAGs as we include complete genomic information of bacterial isolates.

Moreover, as stated in our last response, if the assemblies of the MAGs were incorrect, they would not cluster together in clades, as expected by taxonomy and host identity. Since all samples were assembled and binned separately, one would expect long branches between individual MAGs due to the sample-specific misassemblies, i.e., the binning of genes present in different bacteria/genomes.

Looking at Figure 3, the symbiont of the same bee species scattered among clusters, which is extremely weird. Previous papers showed a clear symbiont diversification among bee species (for example, Kwong et al. Sci. Adv. 2017). The current phylogenetic tree suggests that the variance is higher (or equal) within a bee species than among bee species, which is unlikely to be true. I suggest the authors use all the contigs of microbiota as a reference to build a phylogenetic tree, and compare this tree with the host phylogeny. I also suggest the authors confine the data interruption within symbiont community variance, avoid over stating the results.

Our response: Finding mixed clades of bacteria coming from different hosts may be surprising given some of the conclusions drawn in previous studies and summarized in reviews. However, if one screens the literature carefully, such observations have been made before and are supported by isolated genomes. For example, closely related strains of *Bombilactobacillus* have been isolated from various *Apis* species (Ellegaard et al. 2020). There are also reports of finding *Apis mellifera* strains in bumble bees (Li et al., 2021). Kwong et al. 2017 showed that the community profiles of *A. cerana*, *A. andreniformis*, and *A. florea* are overlapping, suggesting that they must share community members (See Fig. 3C in Kwong et al.). Likewise, Kwong et al. showed in Figure 5 of their paper that several OTUs (clustered at 99%) are present across host species, e.g. for *Lactobacillus Firm5*, *Bifidobacterium*, and *Snodgrassella*. So, while it is true that many clades are host-specific, several community members are shared, or only show signs of host specificity at the strain-level. Hence, the results of our study are not in stark contrast with the previous literature upon careful screening. It is also important to note that the previous studies were based on data with fewer honeybee samples, less breadth or taxonomic depth.

Reviewer #3 (Remarks to the Author):

I appreciate the author's response to my main concern (reply 3.2). It is now clear that all MAGs, including strains, were used for the analysis rather than only dereplicated MAGs at the species level—this approach makes more sense.

Our response: We thank the reviewer.

Thank you also for clarifying the results in Table S6 (reply 3.3) and updating it for better clarity. The addition of tanglegrams to visualize nodes that pass various thresholds (Supplementary Fig. S12) is a valuable supplement, as it more clearly illustrates that “several nodes show congruence but only cover three different host species, with one of them represented by just one or two MAGs while others have several, which can lead to increased R and significant p-values”. I also appreciate the revision to the text, adding a note comparing these findings to observations in hominid microbes.

Our response: We are happy to read that this reviewer values the new addition and agrees with the interpretation of our results. We agree with this reviewer that the comparison to the hominid dataset of Sanders et al. is an important addition because it shows that the signal for co-diversification in social bees is clearly lower than in hominids.

The authors have also addressed all my minor suggestions.

Our response: We thank the reviewer for their help in improving our manuscript.

Reviewer #4 (Remarks to the Author):

In this study, Prasad et al performed shotgun sequencing on the hindgut of multiple bee species. Among other analyses, the authors tested for co-diversification. Due to the high R value cutoff they imposed, they conclude co-diversification did not occur in most species pairs. As part of this analysis, the authors omit significant correlations below .75 (eg, moderate-strong correlation of $.4 < R$ and weak correlation of $.2 < R$). Because of this omission, the authors conclude most nodes do not correspond with co-diversification.

In my previous comments, I noted that inclusion of nodes with an $R < .4$ or $< .2$ would demonstrate that co-diversification is occurring, but is not the sole process driving evolution (ie, host switching may also occur).

I recommended:

1. Fleshing out the visualization of the analysis assessing co-diversification

-Include tanglegrams

-Visualize strong, moderate, and weak correlation

2. Change the conclusion (including the title of the paper) to reflect that co-diversification occurs, but is only a partial driver of evolution.

The authors have addressed point 1, but not point 2.

Our response: We thank the reviewer for their second in-depth revision of our manuscript. In particular, their second comment on the interpretation of our cophylogenetic results encouraged us to think deeply about this result and reframe our thinking around co-diversification. This made us realize that a longer discussion of the interpretation of these results was warranted, and we now acknowledge and discuss the signals of co-diversification that we detect. We have followed the two recommendations made by the reviewer below, namely, update the title and conclusions (see specific changes proposed below). We also respond below in more detail to some of the other comments made by the reviewer.

New title: ***“Symbiont loss and gain predominate over co-diversification in shaping honeybee gut microbiota diversity and function”***

Change in **abstract** L 30-33: *“While we found some evidence of co-diversification between hosts and symbionts, this was not more than expected by chance, and much less pronounced than what has been observed for gut bacteria of hominids and small mammals. Instead, symbiont gains, losses, and replacements emerged as more important factors for honeybees.”*

Change in the **results**, L 258-262: *“In particular, we observed that in some clades, MAGs originating from *A. florea* were sometimes closely related to clades of MAGs originating from *A. andreniformis* and these two host species are also closely related. This is an example where one single internal node is congruent between symbiont and host phylogenies.”*

Change in **discussion** L 497-499: *“While co-diversification, host switches, gains and losses participate to the evolution of the honey bee gut microbiome, host switches, gains and losses seem to be the predominant mechanism.”*

The authors have added tanglegrams as part of Fig S12. In line 232 of the main text, the authors note “Closer inspection of the subtrees from these nodes showing the highest signals of potential co-diversification for each genus showed little congruence with host phylogeny”. However, again, I reach the opposite conclusion looking at these data. Each of the tanglegrams shown have a high R value, indicating a strong correlation between host and symbiont. Further, most of the tanglegrams shown visually depict a high level of concordance between host and symbiont, with very little “tangling” (Frischella has some tangling, but also some concordance). These data support the conclusion that co-diversification occurs in most host-symbiont pairs to an extent, with host-switching also likely contributing to evolution.

Our response: We think the discrepancy between our interpretation and the reviewer’s interpretation stems from a lack of clarity in the manuscript on what is measured with the co-phylogenetic method and visualization we use. We have explained this below and also clarified it in the text. We think this will be valuable for future readers.

Neither the Hommola test nor the tanglegram visualization differentiates between the congruency of symbiont and host BETWEEN vs. WITHIN each host species. Indeed, the Hommola test we use here measures the correlation between the phylogenetic distance of host and symbionts using distances coming from pairs of MAGs from **both** the same and different host species. This test can hence report a high correlation in the following two non-mutually exclusive scenarios of long-term, intimate association between host and symbiont (this was reported before, e.g., by (Moeller et al., 2023; Nishida & Ochman, 2021; Sanders et al., 2023).

A/ at relatively **short** timescales (i.e., **within** host species). This corresponds to cases where a large monophyletic group of MAGs is found in one host species and hence contributes to the phylogenetic congruence signal.

B/ at relatively **long** time scales (i.e., **between** host species). This corresponds to cases where there is phylogenetic congruence between deeper **internal** nodes of the host phylogeny – and this is what we refer to when writing about “co-diversification”. This is rare in our dataset, except in a couple of cases (developed below and now in the manuscript).

Scenario A represents a clear pattern of host specificity – driven by large monophyletic groups - but does not necessarily imply an underlying pattern of co-diversification, while scenario B identifies candidates for co-diversification. Scenario A is very common in our dataset and leads to high effect sizes (and significant p-values) in the Hommola test as well as very low entanglement in the tanglegrams. The archetypical example of this is found for *Frischella* in Supplementary Fig. S12K (and copied below): Hommola R is extremely high ($R=0.82$, $p<.001$), and there is high correspondence between symbiont and host within host species. In this case, all *Frischella* MAGs found in *A. mellifera* form a clade, and this is similar (albeit weaker) for *Frischella* MAGs from *A. cerana* and *A. dorsata*. These closely related MAGs within each clade lead to a large effect size and significant p-value. However, upon closer inspection, it is clear that there is no host internal node congruency (i.e., the MAGs from *A. cerana* are more closely related to the MAGs of *A. dorsata* even though *A. mellifera* and *A. cerana* are the more closely related sister species with *A. dorsata* being more divergent.)

Scenario B is less common in our dataset, but some examples can be found among the clade with larger effect sizes (higher than our strictest thresholds) in the Hommola test. In particular, in several independent bacterial clades (e.g., *Bifidobacterium* (see tree below), *Commensalibacter*, *Bombilactobacillus*), it can be observed that clades of MAGs originating from *A. florea* are closely related to clades of MAGs originating from the found *A. andreniformis* and these two host species are also closely related. This is an example where **ONE** single internal node is congruent between the two phylogenies.

We think making this distinction between scenarios A and B is key: while they both support a long-term, intimate relationship between host and symbiont and both influence the Himmola test, they critically differ in the timescale of this intimate relationship. We have now acknowledged this in the new version of this manuscript.

L 248-258 : “*Closer inspection of the subtrees from nodes showing the highest signals of potential co-diversification for each genus revealed that the Himmola congruency signal was driven by two effects (Supplementary Fig. S12-S13). The first effect originates from strong phylogenetic host specificity but not co-diversification across host species. In this case, distinct clades (=monophyletic groups) of MAGs are found in distinct host species, but the relatedness between these clades does not match host species relatedness (see Frischella clade in Supplementary Fig. S12K and highlighted node in Fig. 3). The second effect originates from congruency between internal nodes of both host and symbiont phylogenies, for example, in some clades of Bifidobacterium, Commensalibacter or Bombilactobacillus (Supplementary Fig. S12-S13 and highlighted nodes in Fig. 3), suggesting ancient co-diversification (across host species).*”

We acknowledge that, in the previous manuscript, we could have highlighted the evidence for scenario B. We did not because the actual congruence pattern is not more frequent than expected *by chance alone*. That being said, we agree that we might have been too stringent, and the fact that the examples of internal node congruence repeatedly involve dwarf honeybees is intriguing. We now acknowledge this in the conclusion and ensure that this is reflected in the new title.

In the results, L 258-262: “*In particular, we observed that in some clades, MAGs originating from A. florea were sometimes closely related to clades of MAGs originating from A. andreniformis and these two host species are also closely related. This is an example where one single internal node is congruent between symbiont and host phylogenies.*”

In the discussion, L 402-416: “*We found that some bacterial clades showed signals of phylogenetic congruence with their host (Supplementary Fig. S12-S13, Fig. 3). However, in most of these cases, the signal was driven by a large number of related MAGs found in a single host species but not by congruence of internal nodes. In other words, clades of MAGs specific to a given host species did not neighbor those from the closest relatives of that host. This pattern of high host phylogenetic specificity has been also reported in other animals and suggests that, while there is an intimate relationship between host and symbionts within host species, this relationship largely breaks down over longer time scales (i.e. across host species). One intriguing exception to this is that in multiple bacteria genera, a clade of MAGs originating from A. florea was sometimes closely related to a clade of MAGs originating from A. andreniformis and these two host species are also closely related. This rare instance of phylogenetic congruence occurs between the two most related host species in our dataset that diverged 6.42 million years ago (Carr, 2023). Along with the strong signal of phylogenetic host specificity, this supports the idea that intimate relationships between host and symbionts are maintained across the divergence of closely related species but are lost over longer timescales due to occasional symbiont loss and gains.*”

Fig S13 depicts tanglegrams with varying levels of R value stringency. The relaxed model, illustrates well that in nodes with weaker correlations, both co-diversification and host switching occur.

Our response: As explained above, these examples seem heavily influenced by scenario A but do not totally reject scenario B (one node is indeed congruent). We also notice now that the example tanglegram shown for the relaxed threshold was not the most representative. We have now provided two more examples that better represent the nodes picked up by the relaxed threshold. The first example that we provided, **Supplementary Fig. S13C.**, represents a node containing two sub-trees, one of which is picked up by the strict signal. While the subtree itself does show a more convincing signal ($R = 0.77$), the node picked up by the relaxed threshold is not congruent to the host phylogeny. To further exemplify this and the effect of large monophyletic clades on the R-value, we have included two more trees (**Supplementary Fig. S13D-E**) where a node (**S13-D**) representing an incongruent tree (fitting neither scenario A nor B) is picked up by relaxing the threshold ($R = 0.26$) and another

node (**S13-E**) containing its tree as a sub-tree is picked up with an even higher R value ($R = 0.496$) primarily driven by a large monophyletic clade of MAGs from *A. dorsata*.

We updated Supplementary Figure S12 accordingly.

The interpretation that host-switching and co-diversification both contribute to evolution are consistent with previous work. Koch et al 2013, for example, shows that both co-diversification and host-switching contribute to evolution in *Snodgrassella* and *Gilliamella*. In this work, no R value cutoff is used for Mantel tests. (R values were found to be .680 and .332 for these genera respectively). Tanglegrams from this study also illustrate a mix of concordance and “tangledness”, similar to in Prasad et al 2024. Koch et al conclude these data, “reject a strict cospeciation of *Snodgrassella* and *Gilliamella* with their bee hosts. The association between these bacteria and bees appears to allow for a considerable degree of host switching... Nevertheless, for both *Snodgrassella* and *Gilliamella*, we found a significant correlation between the 16S rRNA gene bacterial phylogeny and the host phylogeny. The reconciliation analyses furthermore suggest the possibility for a number of cospeciation events.” In other words, the range of strength of correlations they observe point to both co-diversification and host-switching as drivers for evolution. Prasad et al 2024’s data align with those of Koch et al 2013, so the conclusions should be similar.

Our response: We have now modified our conclusion to reflect our results better and to relate it to previous work, especially the seminal work by Koch et al. 2013. As in Koch et al. 2013, we found that host switches, losses and gains are more prevalent than strict co-diversification. This is reflected by the higher inferred rate of switches vs co-speciation in their reconciliation analysis (rate host switches/rate co-speciation events > 1 and up to 2.2 and 3.0 depending on the genera, Koch et al. 2013).

That being said, there are a number of important differences between Koch et al 2013 and our study that make it difficult to directly compare results, including:

- **Single vs multiple node approach:** We used cutoffs because we screened the entire genus phylogenies, while Koch et al. 2013 only used one phylogeny per genus
- **Host sampled:** we analyzed honeybee microbiomes while Koch *et al.*’s dataset was dominated by sequences from bumble bee bacteria. This said, Koch *et al.* noted that the few *Apis* sequences they had included in their analysis sometimes did not form host-specific clades and, hence, were not congruent with the host phylogeny. This is exciting, and future studies on bumble bee bacteria with more resolved datasets should be analyzed to see if there is indeed a difference in the degree of co-diversification between the two groups of bees and their gut symbionts.

This is now clearly acknowledged in the discussion.

L 437-450 : “*These results appear to contradict previous studies that reported evidence for co-diversification between bee gut symbionts and their hosts (Koch et al., 2013; Kwong et al., 2017; Li et al., 2022; Powell et al., 2016). This discrepancy may arise from several factors, such as earlier analyses not employing statistical tests and corrections, not utilizing genome-scale phylogenies or not sampling the full diversity of microbiota, as they relied on isolate genomes. However, even in those earlier studies, it was acknowledged that host switches, symbiont gains, and losses must have occurred, as evidenced by the imperfect congruency between host and bacterial phylogenies. For example, one study that applied statistical methods and found evidence for co-diversification still estimated that the rate of co-speciation was about half the rate of symbiont switches (Koch et al., 2013). Furthermore, this study focused primarily on bumble bee bacteria from the genera Snodgrassella and Gilliamella. It is possible that the signal for co-diversification in gut symbionts of bumble bees may be stronger compared to those from honeybees, a hypothesis that should be tested in future studies using more comparable approaches. In any case, our results markedly contrast with co-diversification findings in mammals, specifically hominids (Arora et al., 2023; Good, 2023; Moeller et al., 2016; Rühlemann et al., 2024; Sanders et al., 2023; Suzuki et al., 2022) and rodents (Sprockett et al., 2025).*”

With this in mind, in their revised manuscript, the authors still do not acknowledge the role that co-diversification plays (in conjunction with host-switching) in evolution.

Our response: We now acknowledge its role in the discussion, conclusion and title (see our detailed replies above).

New title: **“Symbiont loss and gain predominate over co-diversification in shaping honeybee gut microbiota diversity and function”**

Change in **abstract:** “While we found some evidence of co-diversification between hosts and symbionts, this was not more than expected by chance, and much less pronounced than what has been observed for gut bacteria of hominids and small mammals. Instead, symbiont gains, losses, and replacements emerged as more important factors for honeybees.”

Change in **discussion** L 497-499 :” While co-diversification, host switches, gains and losses participate to the evolution of the honey bee gut microbiome, host switches, gains and losses seem to be the predominant mechanism.”

In order to support publication of this manuscript, I recommend:

1. The authors update their conclusion to reflect that both co-diversification and another process, such as host-switching, contribute to species evolution.

Our response: We have updated both the results, discussion and conclusion. We now acknowledge that co-diversification contributed to gut microbiota evolution in honeybees, but come to the conclusion that symbiont gain and loss must have been predominant (see above for quotations).

2. Update their title to reflect this conclusion.

Our response: We have updated our title to: **“Symbiont loss and gain predominate over co-diversification in shaping honeybee gut microbiota diversity and function”**.

This title acknowledges that co-diversification has occurred but that symbiont gain and loss were the dominant processes. We come to this conclusion for several reasons: (1) While many clades are host-specific, the internal node congruency between bacteria and host phylogenies is low (and not more than expected by chance), this suggests that ancient (i.e. across host species) co-diversification is rare. (2) Several clades contain MAGs originating from different honeybee species, suggesting recent/ongoing host switching or overlapping host range. (3) Several bee species are deprived of some of the core symbionts (e.g., *Snodgrassella* and *Gilliamella* in *A. florea* indicating symbiont loss). (4) *A. dorsata* contains several genera not found in the other bees, suggesting ancient loss in all others, or gains in *A. dorsata*. (5) Host-specific symbionts (such as *Dysgonomonas*) contribute to functional differences across honeybee species. Hence, not only the diversity but also the functions of the microbiota are shaped by symbiont gains and losses. Our title reflects these major observations of our study.

References

Arora, J., Buček, A., Hellemans, S., Beránková, T., Romero Arias, J., Fisher, B. L., Clitheroe, C., Brune, A., Kinjo, Y., Šobotník, J., & Bourguignon, T. (2023). Evidence of cospeciation between termites and their gut bacteria on a geological time scale. *Proceedings of the Royal Society B: Biological Sciences*, 290(2001), 20230619. <https://doi.org/10.1098/rspb.2023.0619>

- Carr, S. M. (2023). Multiple mitogenomes indicate Things Fall Apart with Out of Africa or Asia hypotheses for the phylogeographic evolution of Honey Bees (*Apis mellifera*). *Scientific Reports*, 13(1), Article 1. <https://doi.org/10.1038/s41598-023-35937-4>
- de Vienne, D. M., Refrégier, G., López-Villavicencio, M., Tellier, A., Hood, M. E., & Giraud, T. (2013). Cospeciation vs host-shift speciation: Methods for testing, evidence from natural associations and relation to coevolution. *New Phytologist*, 198(2), 347–385. <https://doi.org/10.1111/nph.12150>
- Dismukes, W., Braga, M. P., Hembry, D. H., Heath, T. A., & Landis, M. J. (2022). Cophylogenetic Methods to Untangle the Evolutionary History of Ecological Interactions. *Annual Review of Ecology, Evolution, and Systematics*, 53(Volume 53, 2022), 275–298. <https://doi.org/10.1146/annurev-ecolsys-102320-112823>
- Good, B. H. (2023). *Limited codiversification of the gut microbiota with humans* (p. 2022.10.27.514143). bioRxiv. <https://doi.org/10.1101/2022.10.27.514143>
- Johnson, K. P., Adams, R. J., Page, R. D. M., & Clayton, D. H. (2003). When Do Parasites Fail to Speciate in Response to Host Speciation? *Systematic Biology*, 52(1), 37–47. <https://doi.org/10.1080/10635150390132704>
- Koch, H., Abrol, D. P., Li, J., & Schmid-Hempel, P. (2013). Diversity and evolutionary patterns of bacterial gut associates of corbiculate bees. *Molecular Ecology*, 22(7), 2028–2044. <https://doi.org/10.1111/mec.12209>
- Kwong, W. K., Medina, L. A., Koch, H., Sing, K.-W., Soh, E. J. Y., Ascher, J. S., Jaffé, R., & Moran, N. A. (2017). Dynamic microbiome evolution in social bees. *Science Advances*, 3(3), e1600513. <https://doi.org/10.1126/sciadv.1600513>
- Li, Y., Leonard, S. P., Powell, J. E., & Moran, N. A. (2022). Species divergence in gut-restricted bacteria of social bees. *Proceedings of the National Academy of Sciences*, 119(18), e2115013119. <https://doi.org/10.1073/pnas.2115013119>
- Moeller, A. H., Caro-Quintero, A., Mjungu, D., Georgiev, A. V., Lonsdorf, E. V., Muller, M. N., Pusey, A. E., Peeters, M., Hahn, B. H., & Ochman, H. (2016). Cospeciation of gut microbiota with hominids. *Science*, 353(6297), 380–382. <https://doi.org/10.1126/science.aaf3951>

- Moeller, A. H., Sanders, J. G., Sprockett, D. D., & Landers, A. (2023). Assessing co-diversification in host-associated microbiomes. *Journal of Evolutionary Biology*, *n/a(n/a)*.
<https://doi.org/10.1111/jeb.14221>
- Moran, N. A., Ochman, H., & Hammer, T. J. (2019). Evolutionary and Ecological Consequences of Gut Microbial Communities. *Annual Review of Ecology, Evolution, and Systematics*, *50*(1), 451–475. <https://doi.org/10.1146/annurev-ecolsys-110617-062453>
- Nishida, A. H., & Ochman, H. (2021). Captivity and the co-diversification of great ape microbiomes. *Nature Communications*, *12*(1), 5632. <https://doi.org/10.1038/s41467-021-25732-y>
- Powell, E., Ratnayeke, N., & Moran, N. A. (2016). Strain diversity and host specificity in a specialized gut symbiont of honeybees and bumblebees. *Molecular Ecology*, *25*(18), 4461–4471.
<https://doi.org/10.1111/mec.13787>
- Rühlemann, M. C., Bang, C., Gogarten, J. F., Hermes, B. M., Groussin, M., Waschina, S., Poyet, M., Ulrich, M., Akoua-Koffi, C., Deschner, T., Muyembe-Tamfum, J. J., Robbins, M. M., Surbeck, M., Wittig, R. M., Zuberbühler, K., Baines, J. F., Leendertz, F. H., & Franke, A. (2024). Functional host-specific adaptation of the intestinal microbiome in hominids. *Nature Communications*, *15*(1), Article 1. <https://doi.org/10.1038/s41467-023-44636-7>
- Sanders, J. G., Sprockett, D. D., Li, Y., Mjungu, D., Lonsdorf, E. V., Ndjango, J.-B. N., Georgiev, A. V., Hart, J. A., Sanz, C. M., Morgan, D. B., Peeters, M., Hahn, B. H., & Moeller, A. H. (2023). Widespread extinctions of co-diversified primate gut bacterial symbionts from humans. *Nature Microbiology*, *8*(6), Article 6. <https://doi.org/10.1038/s41564-023-01388-w>
- Sprockett, D. D., Dillard, B. A., Landers, A. A., Sanders, J. G., & Moeller, A. H. (2025). Recent genetic drift in the co-diversified gut bacterial symbionts of laboratory mice. *Nature Communications*, *16*(1), 2218. <https://doi.org/10.1038/s41467-025-57435-z>
- Suzuki, T. A., Fitzstevens, J. L., Schmidt, V. T., Enav, H., Huus, K. E., Mbong Ngwese, M., Griebshammer, A., Pfeleiderer, A., Adegbite, B. R., Zinsou, J. F., Esen, M., Velavan, T. P., Adegnika, A. A., Song, L. H., Spector, T. D., Muehlbauer, A. L., Marchi, N., Kang, H., Maier, L., ... Ley, R. E. (2022). Codiversification of gut microbiota with humans. *Science*, *377*(6612), 1328–1332. <https://doi.org/10.1126/science.abm7759>

Our comments are in blue below. All cited references in the responses are listed at the end of the document. Lines number corresponds to marked-up manuscript.

Reviewer #2 (Remarks to the Author)

A single time point collected samples can't infer symbiont gain or loss. I suggested the authors confine their conclusion to symbiont community dynamics among bee species, which is also interesting. However, the authors insist on drawing this essential conclusion on the evolution of bee symbiont gain and loss, which is substantially overstated from their data

Reply: Our approach to infer the evolution of gut symbionts (*i.e.*, symbiont gain and loss but also co-diversification) is based on comparative phylogenomics. This well-established discipline combines genomics with phylogenetic trees to study the evolutionary history of contemporary species. Most comparative phylogenomic studies are based on what the reviewer refers to as 'single time point collected samples' (*i.e.*, data from only contemporary species). It has been shown - many times - that one can infer the evolutionary history of species from such data (including symbiont gain and loss, co-diversification etc.). Comparative phylogenomic approaches have been thoroughly reviewed (e.g. in Dismukes *et al.* 2022, D. M. de Vienne *et al.* 2013, Groussin *et al.* 2020) and applied in hundreds of different studies, including many microbiome studies in which genomes of individual microbes have been reconstructed from metagenomes: Sprockett *et al.* Nat Commun 2025; Sanders *et al.* Nature Microbiol 2023; Rühlemann *et al.* Nature Commun 2024. One of the most influential papers of the last decade, Spang *et al.* Nature 2015, which has changed our view on the tree of life, has applied exactly this approach.

Please see this review on methods how to infer co-speciation and host shifts in which the authors write: "Macro-evolutionary aspects of host-parasite associations cannot be observed within the lifespan of a researcher. Methods for inferring the effects of interactions have thus been developed based on comparisons of the phylogenies of the interacting species." (D. M. de Vienne *et al.*, New Phytologist, 2013)

Previously, this reviewer raised concerns about our specific approach to reconstructing genomes from metagenomes, worrying about the quality of the reconstructed genomes and hence our ability to infer robust phylogenetic trees based on that data. This was a fair point of critique, which we replied to in our previous response. In short, we have shown that the genomes in our study are reconstructed in the same way as in many other studies in the field, and we have validated these reconstructed genomes by showing that they are highly similar (in terms of phylogenetic signals) to reference genomes of isolated bacteria. The reviewer did not comment on these points further, so we assume they are convinced of our approach.

Reviewer #4 (Remarks to the Author):

In my previous review, I recommended that the authors update their conclusion and title to reflect that both co-diversification and another process, such as host-switching, contribute to species evolution. Despite some minor changes in wording, the authors still conclude that co-diversification occurs "not more than expected by chance" and that host-switching predominates.

Reply: In our understanding, we had updated our manuscript according to the reviewer's comments, as our title and conclusion were updated to reflect that both processes are at play. To avoid any confusion from the title, we have now decided to make it broader and more general, and remove any reference to gains, losses, or co-diversification. We thus propose: "**Evolution of Gut Microbiota Across Honeybee Species Revealed by Comparative Metagenomics**"

Critically, this more general title better reflects the entire study, which is not restricted to gain/losses of symbionts and co-diversification but also encompasses community-level specificity (e.g., phylosymbiosis), gene content analysis and functional analysis.

However, we cannot ignore the results of the co-phylogenetic analysis that is based on a previously published quantitative statistical framework (Sanders et al. *Nature Microbiol* 2023, Moeller et al. *J. Evo. Bio.*, Sprockett et al *Nat Commun* 2025). The reviewer did not comment on or criticize this quantitative approach in their last round of comments, so we assume they consider it appropriate. We find only 15 of the 192 tested nodes across the phylogenies to show a signal of phylogenetic congruence, which is a small fraction of the tested nodes (see updated Fig. 3B, in which we have labelled all significant nodes with a black filled circle). Moreover, this analysis shows that the overall signal we find occurs not more than expected by chance (see Fig. 3C, “observed data” vs “simulated data” per genus phylogeny), despite examples of congruence. It would be bad scientific practice to ignore these results which based on an unbiased, quantitative, and statistical approach applied in previous studies. Nevertheless, we have removed all statements in which we conclude that symbiont gain and loss dominated over co-diversification. Specifically, we have...:

- 1/ ...removed the “*not more than expected by chance*” and the claim that “*symbiont gains, losses, and replacements are more important factors for honeybees than co-diversification*” from the abstract. L36-38 in the marked-up manuscript.
- 2/ ... removed a similar statement at the end of the introduction, we now reads: “... *the evolution of gut microbiota-host interactions in honeybees has been shaped by symbiont loss and gain, and co-diversification*”. L154-156 in the marked-up manuscript.
- 3/ ...specified “*as determined by our permutation test*” when mentioning “*not more than expected by chance*”. L307 in the marked-up manuscript.
- 4/ ...suggested that further studies with more bee species and more isolate genomes should be conducted to validate our results. L585-587 in the marked-up manuscript.
- 5/ ...streamlined the corresponding results section and discussion to report our results more clearly and substantiate our claims about symbiont gain and loss. L259-325 and L525-539 in the marked-up manuscript.
- 6/ ...updated Figure 3 (L1036 in the marked-up manuscript), which now shows all nine genus-level phylogenies with all nodes with significant signal of topological congruence highlighted and with the corresponding subtrees shown in Supplementary Figure S13 (previously Supplementary Figure S12). In addition, we have provided interactive versions of all trees and subtrees in iTOL to explore the data: <https://itol.embl.de/shared/Aiswarya> .

We further would like to emphasize that our study and conclusions about symbiont gain and loss are not only based on the comparative phylogenetic analysis of host and bacteria, but also on the fact that several bacterial genera (and functional gene content) are restricted to a subset of the bees: *Dysgonomonas* and *Apibacter* mostly in *A. cerana* and *A. dorsata*, *Snodgrassella* missing or rare in *A. dorsata* and *A. florea*). These genera (and associated functional gene content) must hence have been lost or gained in some bee species but not in others. Our title and conclusions reflect these results as well.

The authors highlight that the partial lack of concordance between *Frischella* and host phylogeny exemplifies that host-switching predominates in the bee gut microbiome. However, it appears that these data show that the evidence for co-diversification in *Frischella* is simply not as strong as in the other bacterial clades.

Reply: We thank the reviewer for this in-depth comment of fig S12.

We apologize that the reviewer feels that we used *Frischella* to substantiate our claim that “*host-switching predominates in the bee gut microbiome*”, as this was not our intention. We intended to use

the *Frischella* case to exemplify that, despite a very high signal as detected by the method we used, there is no “internal” phylogenetic branch congruence between clades of host-specific symbionts and the host phylogeny, but only phylogenetic clustering by host species (see our detailed explanation in the previous round of review and L294-298 and L313-314 in the marked-up manuscript). As the reviewer did not comment on this important point of our approach and statistical framework (and associated modification in the previous manuscript), we must assume they do not object to it. In fact, besides *Frischella*, there is another example among the subtrees with a significant signal, *Bombilactobacillus* (Supplementary Fig S13F), in which case the signal must also primarily come from the host-specific clustering of MAGs rather than topological congruence between internal branches of the symbiont phylogeny and the host because this tree only includes two clades of host-specific MAGs (from *A. cerana* and *A. andreniformis*; with a single MAG from *A. dorsata* clustering with divergent MAGs of *A. cerana*). We have completely revised this results section as described in our reply to the next point (L259-325 in the marked-up manuscript).

Furthermore, the *Frischella* example seems to be the exception to the rest of the data – all of the other bacterial clades in the Fig S12 tanglegrams (*Bartonella*, *Bifidobacterium*, *Bombilactobacillus*, *Dysgonomonas*, *Commensalibacter*, *Lactobacillus*, *Snodgrassella*) show evidence for co-diversification, based on congruence between host and bacterial branches. The data presented, therefore, do not seem to support the conclusion that host-switching predominates over co-diversification. In fact, the data (Fig S12, tanglegrams) lead to the opposite conclusion in my view -- that co-diversification is common and host-switching is the rarity.

Reply: We thank the reviewer for this in-depth comment of Fig. S12, but we politely disagree with their conclusions that *Frischella* is the exception (see comment above) and that our phylogenetic trees show that co-diversification is common and host switching is rare. In the following, we provide five lines of evidence to substantiate our conclusions that symbiont gain and loss must have been an important process based on both our quantitative statistical framework and the visual inspection of the trees. We also explain how we have revised the manuscript to provide a more balanced view and interpretation on our results and the broader context:

1. Supplementary Fig S12 (now Supplementary Figure S13) only shows the 13 nodes (of a total of 192 tested nodes), which exhibited significant co-phylogenetic signals given our statistical analysis and cutoffs ($R > 0.75$, and $p < 0.01$). All other nodes of the trees of Fig. 3 are not shown in Supplementary Figure S13 and showed no statistically significant signs of co-diversification given our thresholds (thresholds were justified in our previous reply and are in accordance with previous publications). To make these points clear, we have revised the corresponding results section (L259-325 in the marked-up manuscript). Specifically, we first justify our approach and strict cutoffs and then state how many nodes showed signal of phylogenetic congruence with the host (only 13 of 192 nodes). These significant nodes are now depicted with a black filled circle in Figure 3B. To aid comparison, we included a schematic in Figure 3A illustrating how host and symbiont phylogenies would appear in a case of full congruence, highlighting how factors such as internal branch topology and host-specific clustering contribute to congruence signals. This allows readers to directly compare the schematic with the nine genus-level phylogenies of the gut symbionts in Fig. 3B. Based on our visual inspection, we see little evidence for such congruence between clades of host-specific symbionts across the phylogenetic trees of the nine genera shown in Fig. 3B, except for the few nodes that were picked up by our statistical approach. Therefore, we come to the conclusion - based on both our statistical analysis and our visual inspection of the phylogenies – that only a few nodes show topological congruence with the host tree despite the presence of many host-specific clades.

2. Several clades across the nine phylogenies contain MAGs from different host species, indicating recent host switches or ongoing sharing of the same bacteria across host species. On L272 of the marked-up manuscript, we state that ‘only’ 24-47% of all species nodes are host-specific, which

means that many species are detected in more than one host, and hence are shared between different hosts, or have been subjected to relatively recent host switches.

3. In cases of co-diversification, one would expect a rough correlation between host divergence times and the branch lengths separating the corresponding symbiont clades. Based on careful visual inspection (only qualitative, no quantitative analysis) we conclude that this is indeed the case for several subtrees showing significant signal of congruence. Specifically, the divergence of the host-specific symbiont lineages in two subtrees of *Bifidobacterium*, one subtree of *Snodgrassella*, and one subtree of *Commensalibacter* seems to roughly correspond to the relative divergence of the hosts. We specifically mention these as further evidence for co-diversification in the revised results section on L308-312 of the marked-up manuscript (and highlight them in Supplementary Figure S13A-D). However, for other significant nodes, the divergence observed between MAGs of symbionts from different bee species does not appear to align well with host divergence. **Example 1:** In Supplementary Fig S13J – *Bartonella*, branch lengths between MAGs of *A. mellifera* and *A. cerana* (which diverged 6-8 mya) are extremely short (about 10x shorter) compared to the very long branch separating them from the clade of MAGs of *A. dorsata* (which diverged 10-20 mya from the other two bee species). In addition, the MAG of *A. cerana* clusters with a MAG of *A. andreniformis* with little to no divergence between the two, despite the fact that these two bee species have also split about 10 mya. **Example 2:** Clades of MAGs from *A. florea* and *A. andreniformis* are sister to each other in several significant subtrees, which is congruent with the monophyly of the two bee species. However, in some cases, the two clades of MAGs are highly divergent from each other (Supplementary Fig S13A, B, D), while in other cases, there is little to no divergence between the two (Supplementary Fig S13H and I). The two bee species separated about 6 mya. It seems unlikely that some bacteria have not diverged at all, while others have diverged into deep-branching species over the same period of time. Therefore, even in the 13 cases where our analysis detects significant congruence, these patterns cannot always be attributed to co-diversification (because of inconsistent branch lengths, or lack of congruence between host and symbiont topology at internal nodes (see above for our reply to the case of *Frischella*). Hence, in these cases symbiont gains and losses must have been involved. We have mentioned these observations in the results section on L312-316 of the marked-up manuscript.

4. There is not a single case where we find congruence of the symbiont tree across all five host species. All significant nodes included only three to four host species. If co-diversification was the predominant process in these subtrees, the only explanation why we don't observe such relationships across all five host species is that some of these symbiont lineages were lost or gained in a subset of these bees. So, even for the few nodes where we detected significant signal for co-diversification, symbiont gain and loss events must have occurred.

5. Finally, the fact that several 'core' genera of bee gut symbionts are missing or nearly missing in some bee species (see our reply to the first point of this reviewer) is further evidence that gut microbiota members prevalent in some honeybee species must have been lost in other honeybee species, or gained after the bee species have diverged from each other. This is further evidence for a dynamic evolution of the bee gut microbiota, shaped by symbiont gains and losses. Similar observations have been made for prominent gut microbiota members in stingless bees (Cerqueira et al. ISME 2021) and similar processes have already been proposed to also shape the microbiota of honey bees (Kwong et al Sci Adv).

Given these five points, we conclude that overall our analyses show more evidence for host switching and symbiont gain and loss than co-diversification. However, we do acknowledge - as already mentioned in our previous revision - that co-diversification does occur. We have updated the corresponding results section L259-325 to clarify these observations in more detail. We have also updated the corresponding discussion section (L525-539 in the marked-up manuscript) to more systematically list the different pieces of evidence on the basis of which we conclude that host-

switching must have been an important process in the evolution of the honey bee gut microbiota. Finally, we provide a formal definition of co-diversification in the introduction (L62-67 in the marked-up manuscript), which together with the schematic in Fig. 3A should clarify what we define as co-diversification and how we can detect it. Importantly, as explained above, we have decided to tone down our abstract and change the title. We no longer claim that host-switching is more important than co-diversification, but we only highlight that co-diversification signals are lower in mammals. We hope this gives a more balanced view of our results and acknowledges more clearly that co-diversification has occurred.

Reviewer #3 (Remarks to the Author):

I appreciate the authors' efforts to address reviewer concerns. The revised title and removal of earlier overstated claims regarding the predominance of host switching result in a more balanced interpretation of their findings.

Reply: We are pleased to note that this reviewer agrees with the way we have addressed the earlier concerns of reviewer #4.

Overall, the manuscript (as well as the responses to the reviewers) is technically sound and includes appropriate analyses. The conclusions are now more cautiously framed, as in L116: "the evolution of specialized gut microbiota-host interactions in honeybees has been shaped by symbiont loss and gain, and co-diversification." However, the primary contribution of this conclusion remains largely confirmatory of previously described patterns in bee microbiome evolution.

Reply: We are happy to read that this reviewer found our study technically sound. In our study, we formally tested for different evolutionary processes that underlie phylogenetic patterns. However, we politely disagree that our conclusions remain largely confirmatory. While previous studies have proposed these processes—generally assuming predominant co-diversification—they have not been formally tested in honeybees and the strength of the co-diversification signal has never been compared to other systems, such as primate gut microbiomes. In addition, our study reveals that there can be significant variations in specificity among bacterial gut symbionts, even between closely related species. This has not been shown in previous studies and raises important questions about the various factors that determine the specificity of gut symbionts (e.g., host vs bacterial factors). Therefore, we think that our paper's contribution clearly goes beyond confirming previously described patterns and will guide future research efforts.

Limited novelty: While the analyses are carefully conducted and the recovery of many novel *A. dorsata* MAGs is valuable, the study's main conclusions largely reinforce what has already been proposed. As the authors themselves acknowledge: "Similar observations have been made for prominent gut microbiota members in stingless bees (Cerqueira et al. ISME 2021) and similar processes have already been proposed to also shape the microbiota of honey bees (Kwong et al. Sci Adv)." Their central finding (L369) that "our findings point to an important role for symbiont gain and loss, as well as host-switching, in shaping these specialized gut communities" essentially confirms established dynamics.

Reply: As noted in our previous reply, we respectfully disagree with the reviewer's assessment of our study's novelty, though we appreciate the opportunity to clarify our contribution. While similar evolutionary processes have been proposed in earlier work, our study is, to our knowledge, the first to formally and quantitatively test these processes in the honeybee gut microbiota. Previous studies have suggested co-diversification between honeybees and their gut symbionts, but have not applied the recent analytical framework that allows comparison across gut microbiota systems. Likewise, host switching, symbiont gain, and loss have been inferred from phylogenetic patterns, yet not rigorously tested. Our study addresses this gap by applying statistical tests for codiversification across a broader dataset than used previously, thus providing the first quantitative insights into hypotheses that until now have remained speculative. We have slightly revised the Discussion to better emphasize this key aspect of our study, which supports the novelty of our contribution (L370). The sentences reads as follows: "*While it is generally assumed that the honeybee microbiota has co-diversified with their hosts - with occasional interruptions - previous studies have not applied recent analytical frameworks that allow comparison across gut microbiota systems to rigorously test this assumption (Li et al. 2022, Kwong et al 2017). Using our analytical approach, we found limited evidence for strict co-diversification between gut bacteria and their hosts across the five analyzed Apis species.*"

Furthermore, our study also provide evidence of significant variation in host specificity among closely related bacterial symbionts (i.e. distribution across closely related hosts in nature/host range). This has not been shown before and highlights the complexity of host-microbe associations in this system. By identifying and analyzing these patterns, our study opens up new directions for investigating the ecological and evolutionary drivers of gut symbiont specificity. Our findings hence extend beyond confirming earlier hypotheses and offer valuable new insights for the field.

Co-diversification Analysis: Although I must note that host–microbiome co-evolution using such methods is not my specific area of expertise, the analyses and explanations provided by the authors seem appropriate and (mostly) carefully interpreted. However, the use of an $r > 0.75$ cutoff to define co-diversifying clades may be overly conservative — a point raised by reviewer #4, which I agree with. Many clades show moderate levels of phylogenetic congruence (52 out of 103 with $r > 0.4$, Table S5), which may indicate real co-diversification signals

that are being overlooked under strict thresholds. The authors argue, for example, that "Low r-values accompanied by significant (low) p-values can arise due to a large number of MAGs." However, this explanation may be less relevant here, given that their dataset includes only 1,959 medium+high-quality MAGs, in contrast to the much larger dataset in Sanders et al. (9,460 high quality MAGs). Also, in sympatric *Apis* species (sharing environmental microbial pools) that also have significantly different life cycles than the other systems being compared, as mammals where such analyses were also done, detecting clear co-diversification signals may be inherently more difficult and other cutoffs maybe should be considered.

Reply:

We appreciate the reviewer's thoughtful comments and their agreement that the analyses and interpretations are generally appropriate. We agree that using an $r > 0.75$ threshold is conservative. As the reviewer notes, many clades with moderate congruence (e.g., $r > 0.4$) may still show biologically meaningful patterns of co-diversification. We have now included a figure illustrating how the number of significant trees varies as a function of the R-value threshold.

This allows readers to visualize the distribution of congruence values and better assess the potential for co-diversification across a broader range of thresholds. Additionally, we have provided R values in the names of the trees in iTOL that enable readers to interactively explore and sort subtrees based on these values. From here it becomes clear that lower R values include trees with few signs of congruence often containing large host-specific clades but no other signs of topological congruence.

Regarding our comment about significant p-values in trees with low R values: our point was that these patterns can sometimes arise when there is a large number of MAGs from a few host-specific clades, which inflates the apparent significance even if topological congruence is limited. We acknowledge that the overall number of MAGs in our dataset is smaller than in Sanders et al., but the clustering of MAGs within certain host-specific lineages remains relevant in our case. This reasoning was explained in greater detail in the previous round of reviews.

Most importantly, regardless of the threshold applied (we report results for $R > 0$ and $R > 0.75$ in Supplementary Table S6), the number of codiversifying nodes observed in our data is **comparable to that expected under the null model, which is based on trees with randomized tip labels**. In other words, irrespective of the chosen R-value cutoff, **the number of significant associations in our actual dataset does not exceed what would be expected by chance**. By contrast, in the study by Sanders et al., despite similarly high false discovery rates, they observed approximately 10 times more significant nodes in their actual data compared to the null, indicating a statistically significant signal for codiversification.

We have slightly revised the Abstract (L30) and the Discussion (L374) to better emphasize this important point of our analysis.

Abstract from L30: "*While we found some signal of co-diversification between hosts and symbionts, this was not more than expected by chance and was much less pronounced than what has been observed for gut bacteria of hominids and small mammals.*"

Discussion from L374: "*We found limited evidence for strict co-diversification between gut bacteria and their hosts across the five analyzed *Apis* species. Although we applied relatively stringent thresholds (i.e., high R values) to detect signals of co-diversification, the second-order permutation analysis revealed that the number of*

significant nodes did not exceed what would be expected by chance independent of the chosen R-value cutoff. Rather than supporting widespread co-diversification between gut bacteria and honeybee hosts, our findings highlight the importance of symbiont gain and loss, as well as host switching, in shaping these specialized gut communities.”

Interpretation: I appreciated the detailed view of co-diversifying clades in Supplementary Figure S12 and S13. It effectively illustrates cases where "there is no internal phylogenetic branch congruence between clades of host-specific symbionts and the host phylogeny, although phylogenetic clustering by host species", but also reveal cases where moderate phylogenetic congruence signals may warrant closer consideration (e.g., Figure S12B). Also, "no internal phylogenetic branch congruence" could reflect differences in evolutionary rates rather than definitively pointing to host switching, and this possibility should be more carefully acknowledged in the discussion. Additionally, while maybe this was not the main question and is not testable using the co-diversification methods applied here (which require the symbiont to be present in multiple host species), the consistent restriction of certain taxa to a single *Apis* host (e.g. *Dysgonomonas* mostly in *A. dorsata*) could reflect a long-term host association and raise questions about possible host-specific coevolution.

Reply: We are happy to read that the reviewer agrees with the way we interpret and discuss our results. Our method did not consider branch lengths (but only topology) for correlating host and bacterial phylogenies. However, we have included one more sentence in the Discussion on line 392 of the revised (mark-up) manuscript to discuss the fact that differences in branch length have been observed and could be further evidence for host switchg, but may also reflect differents in rates of evolution. It reads as follows:

Discussion L392:” *Fifth, internal branch lengths varied substantially among splits of symbionts associated with the same hosts, suggesting that these divergence events occurred at different points in evolutionary time, although they could also reflect differences in evolutionary rates.”*